# DNA methylation reprogramming during seed development and its functional relevance in seed size/weight determination in chickpea

Mohan Singh Rajkumar[1], Khushboo Gupta[2], Niraj Kumar Khemka[1], Rohini Garg ![ORCID] [2✉] & Mukesh Jain ![ORCID] [1,3✉]

Seed development is orchestrated via complex gene regulatory networks and pathways. Epigenetic factors may also govern seed development and seed size/weight. Here, we analyzed DNA methylation in a large-seeded chickpea cultivar (JGK 3) during seed development stages. Progressive gain of CHH context DNA methylation in transposable elements (TEs) and higher frequency of small RNAs in hypermethylated TEs during seed development suggested a role of the RNA-dependent DNA methylation pathway. Frequency of intragenic TEs was higher in CHH context differentially methylated region (DMR) associated differentially expressed genes (DEGs). CG context hyper/hypomethylation within the gene body was observed for most of DMR-associated DEGs in JGK 3 as compared to small-seeded chickpea cultivar (Himchana 1). We identified candidate genes involved in seed size/weight determination exhibiting CG context hypermethylation within the gene body and higher expression in JGK 3. This study provides insights into the role of DNA methylation in seed development and seed size/weight determination in chickpea.

[1] School of Computational and Integrative Sciences, Jawaharlal Nehru University, New Delhi 110067, India. [2] Department of Life Sciences, School of Natural Sciences, Shiv Nadar University, Gautam Buddha Nagar, Uttar Pradesh 201314, India. [3] National Institute of Plant Genome Research (NIPGR), Aruna Asaf Ali Marg, New Delhi 110067, India. ✉email: rohini.garg@snu.edu.in; mjain@jnu.ac.in

Seeds provide bulk of human and animal nutrition and is important for continuation of next generation. Seed size/weight is one of the most desirable traits to fulfill ever-increasing demand of food supply. To understand the mechanisms involved in seed development, gene regulatory networks and their associated pathways have been identified in a few model/crop plants[1–3]. A crucial role of transcription factors and hormonal signaling in seed development and seed size/weight determination have been revealed[4–7].

Epigenetic modifications control reorganization of chromatin architecture to determine euchromatic or heterochromatic regions driven by internal and/or environmental cues in Arabidopsis[8,9]. DNA methylation and histone modifications are the most frequently found epigenetic marks[10,11]. In plants, DNA methylation occurs in three different sequence contexts; symmetric CG and CHG, and asymmetric CHH contexts, where H refers any nucleotide except guanine. DNA methylation in CG context is mediated by DNA Methyltransferase 1 (MET1) in the newly formed DNA strand after each round of DNA replication[12,13]. Chromomethylase 3 (CMT3) marks DNA methylation in CHG context[14,15] and DNA methylation in CHH context is established via small RNAs by recruiting Domains rearranged methyltransferase 2 (DRM2)[16,17]. Another enzyme, Chromomethylase 2 (CMT2) catalyzes DNA methylation in CHH context in highly condensed heterochromatic regions[18,19]. DNA methylation causes silencing of transposable elements (TEs), chromatin reorganization and regulation of imprinted and/or other protein-coding genes[17,20]. The epigenetic marks are dynamic and their massive reprogramming has been observed during gametogenesis and early embryo development[17,21–24]. The primary purpose of reprogramming is to protect the genome in gametes and embryo via small RNA-guided repression of TEs[17,21–24]. Genome protection via methylation of TEs throughout seed development in Arabidopsis and soybean has also been reported[25–27].

Epigenetic regulation of protein-coding genes and processes involved in seed development and seed size/weight determination is largely unknown. Global loss of methylation in CG context due to knockout of MET1 resulted in improper embryo development, reduced/loss of seed viability in Arabidopsis[28] and severe necrotic lesions in rice[29]. Interestingly, loss of methylation in CHG and CHH contexts was not found to affect seed development and seed viability in Arabidopsis and no significant role of DNA methylation in regulation of important genes and processes involved in seed development was suggested[26]. Another study showed that most of important genes involved in seed development were located in constitutively unmethylated regions and did not show differential methylation[30]. Despite no significant correlation between differential methylation and differential gene expression during seed development at global level, a body of evidences have demonstrated regulation of imprinted genes in allele-specific manner especially in endosperm[31–33] and determination of seed size/weight in next generations via genomic imprinting[32].

In this study, we sought to understand epigenetic regulation of seed development and seed size/weight determination in chickpea. We analyzed DNA methylation at single base resolution in different sequence contexts at various stages of seed development in a large-seeded (JGK 3) chickpea cultivar. The differentially methylated regions (DMRs) during seed development were identified and their influence on differential gene expression was interrogated. We analyzed small RNA sequencing data to reveal the role of RNA-dependent DNA methylation (RdDM) pathway in TE methylation. The impact of methylation status of TEs on expression of the associated/proximal protein-coding genes was investigated. The possible role of intragenic TEs in controlling differential methylation and differential expression was revealed.

In addition, we compared the DNA methylomes of JGK 3 with a small-seeded cultivar (Himchana 1) at late-embryogenesis and mid-maturation stages of seed development. Putative differentially methylated candidate genes that might determine seed size/weight were identified. Together, this study provides insights into role of DNA methylation during seed development and seed size/weight determination in chickpea.

## Results

**DNA methylation profiling during seed development.** We performed bisulphite sequencing of the genomic DNA isolated from five successive stages of seed development in a large-seeded cultivar, JGK 3 (100 seed weight of 53.3 ± 1.48 g). We analyzed early-embryogenesis (S1), mid-embryogenesis (S2), late-embryogenesis (S3), mid-maturation (S5), and late-maturation (S7) stages (Fig. 1a), representing important developmental events that occur during seed development as described earlier[3]. About 112–132 million high-quality read pairs were generated for each sample (Supplementary Table 1). About 54–69 million read pairs mapped uniquely that covered 87–89% of the chickpea genome. To verify the efficiency of bisulphite conversion, high-quality reads for each sample were mapped on the chickpea chloroplast genome too. At the most, only 0.006% read pairs mapped to the chloroplast genome (Supplementary Table 1), which confirmed high efficiency of bisulphite conversion. Methylcytosines were identified in CG, CHG and CHH contexts at all the stages of seed development analyzed. A large fraction of cytosines in CG (49.3–58.7%) and CHG (38.3–42.8%) contexts were methylated, but a smaller fraction of cytosines in CHH context (3.9–13.4%) were methylated (Fig. 1b; Supplementary Dataset 1a). The average methylation level of methylcytosines was also much higher in CG (92.2–93.4%) and CHG (83.5–88.8%) contexts as compared to CHH context (34.6–49.8%) (Fig. 1c; Supplementary Fig. 1), which is in agreement with previous studies[25–27]. No significant methylation level variations were observed between forward and reverse strands in the chickpea genome (Supplementary Fig. 2), as reported in other plants too[34,35].

To examine methylation status of all the annotated protein-coding genes at different stages of seed development, we analyzed methylation level within their gene body and flanking regions in all the three sequence contexts (Fig. 1d; Supplementary Dataset 1b). Substantially higher methylation level in CG context was detected within gene body at all the stages of seed development. The CG context methylation level was found to be reduced within gene body and flanking regions from S1 to S5 stages. Likewise, decreased methylation level was observed in CHG context in the flanking regions from S1 to S5 stages, but no remarkable difference was observed within gene body region. Interestingly, progressive gain of DNA methylation in CHH context was observed during seed development (Fig. 1d). The gain of methylation was highest at the S7 stage especially in the flanking regions. Notably, irrespective of DNA methylation in different sequence contexts and stages of seed development, methylation level at gene ends was less, which may be to avoid methylation at/around transcription start site and transcription termination site that can repress gene expression.

**Influence of DNA methylation on gene expression.** To examine influence of DNA methylation on gene expression, all the protein-coding chickpea genes were categorized into different sets based on their expression levels ranging from non-expressed genes to genes expressed at highest level at all the stages of seed development. Methylation status of all these sets of genes was analyzed. Antagonistic relationship of DNA methylation level in

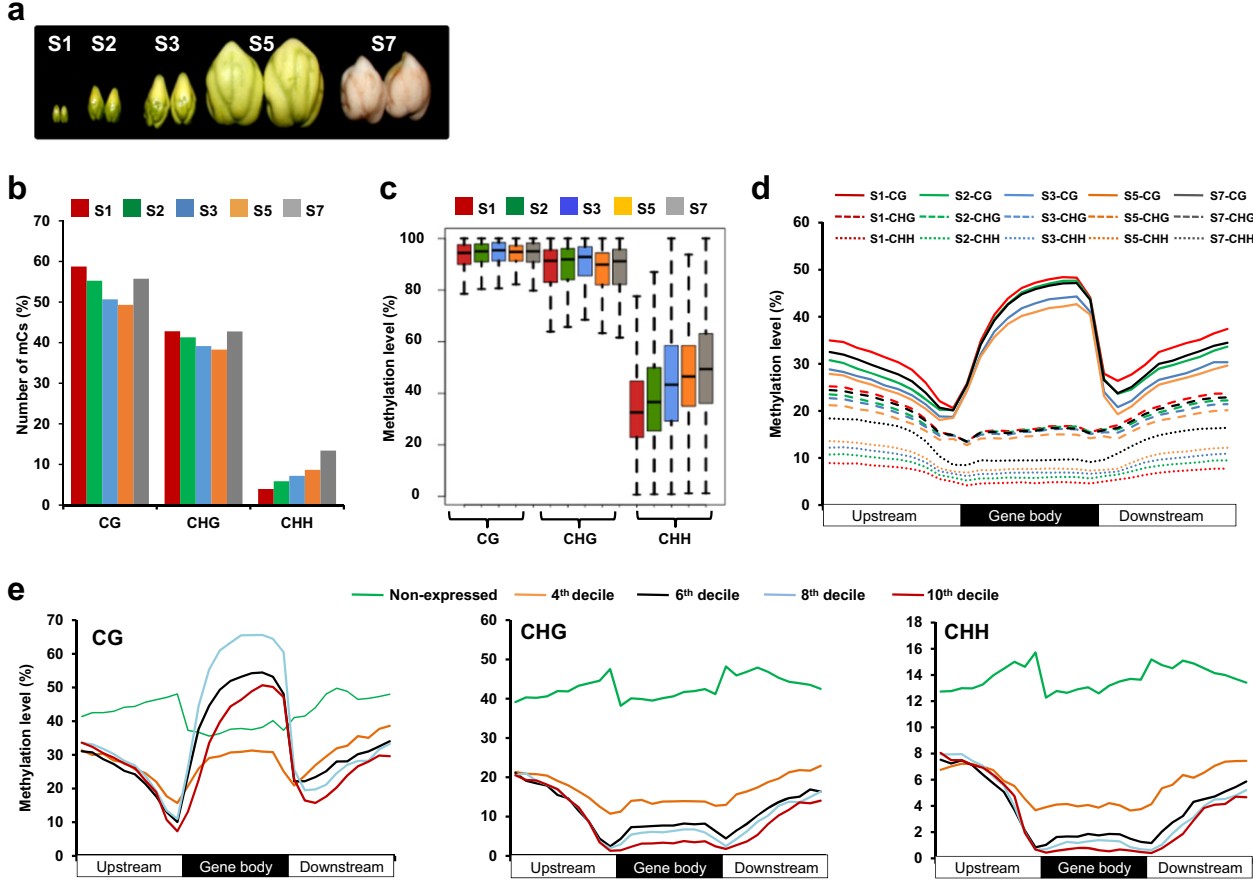

**Fig. 1 Whole genome DNA methylation profiling and influence of DNA methylation on gene expression during seed development. a** Different stages of seed development, including early-embryogenesis (S1), mid-embryogenesis (S2), late-embryogenesis (S3), mid-maturation (S5), and late-maturation (S7) in JGK 3 cultivar[3] used for bisulphite sequencing are shown. **b** Percentage of methylcytosines (mCs) in different sequence contexts (CG, CHG, and CHH) at different stages of seed development is shown in bar graph. **c** Methylation level at individual mC in different sequence contexts at different stages of seed development is shown via boxplot. **d** Methylation level within gene body and 2 kb flanking (upstream and downstream) regions in different sequence contexts for all the protein-coding genes at different stages of seed development is shown. **e** Methylation level within gene body and 2 kb flanking regions in different sequence contexts for the gene sets that are expressed at different levels, including non-expressed (2nd decile), low (4th decile), moderate (6th decile), high (8th decile), and highest (10th decile), at S1 stage is shown. The data for other stages of seed development are given in Supplementary Fig. 3. Each region was divided into 10 bins of equal size and normalized methylation level for the respective set of genes in each bin is shown in the line graphs in **d** and **e**.

CHG and CHH contexts, and gene expression level was observed at all the stages of seed development analyzed (Fig. 1e; Supplementary Fig. 3; Supplementary Dataset 1c). The non-expressed genes and genes expressed at low levels showed higher DNA methylation level at their ends, suggesting that DNA methylation at gene ends can repress gene expression. Interestingly, genes expressed at moderate and high levels were highly methylated in CG context within their gene body (Fig. 1e; Supplementary Fig. 3). However, CG methylation at gene ends showed antagonistic correlation with gene expression levels. The average methylation level in different sequence contexts and genic regions was found to be within the interquartile range (Supplementary Fig. 4), suggesting that the observed methylation patterns were not due to a large change in methylation for a small number of genes or vice-versa. Increasing evidences suggest that DNA methylation in CG context within gene body is correlated with higher gene expression levels[36,37].

**Impact of DNA methylation dynamics on gene expression.** To understand methylation dynamics during seed development, we identified DMRs between successive stages of seed development. A total of 8018–9809 DMRs in all the sequence contexts associated with 3029–3888 genes were identified between successive stages of seed development (S1/S2, S2/S3, S3/S5, and S5/S7) (Fig. 2a, b; Supplementary Dataset 2). These genes were referred as DMR-associated (hypermethylated/hypomethylated) genes hereafter. Most of differential methylation was represented by hypermethylation in CHH context during all the successive stage transitions (Fig. 2a, b). About 97% and 90% of the total DMR-associated genes in S1/S2 and S2/S3 comparisons, respectively, were hypermethylated in CHH context. However, several genes showed progressive hyper/hypomethylation in CG and CHG contexts too at the later stages of seed development. In S3/S5 comparison, the number of hypomethylated genes (81.7%) was much higher than the number of hypermethylated genes (18.3%) in CHG context. Likewise, the number of hypomethylated genes (57.9%) was marginally higher than hypermethylated genes (42.1%) in CG context in S5/S7 comparison, suggesting differential regulation of DNA methylation in CG and CHG contexts at S3/S5 and S5/S7 stage transitions during seed development (Fig. 2a, b). The methylation level differences were also much higher in CG and CHG contexts during S3/S5 and S5/S7 transitions as compared to previous stage transitions (Supplementary Fig. 5). Gene ontology (GO) analysis of hyper and/or

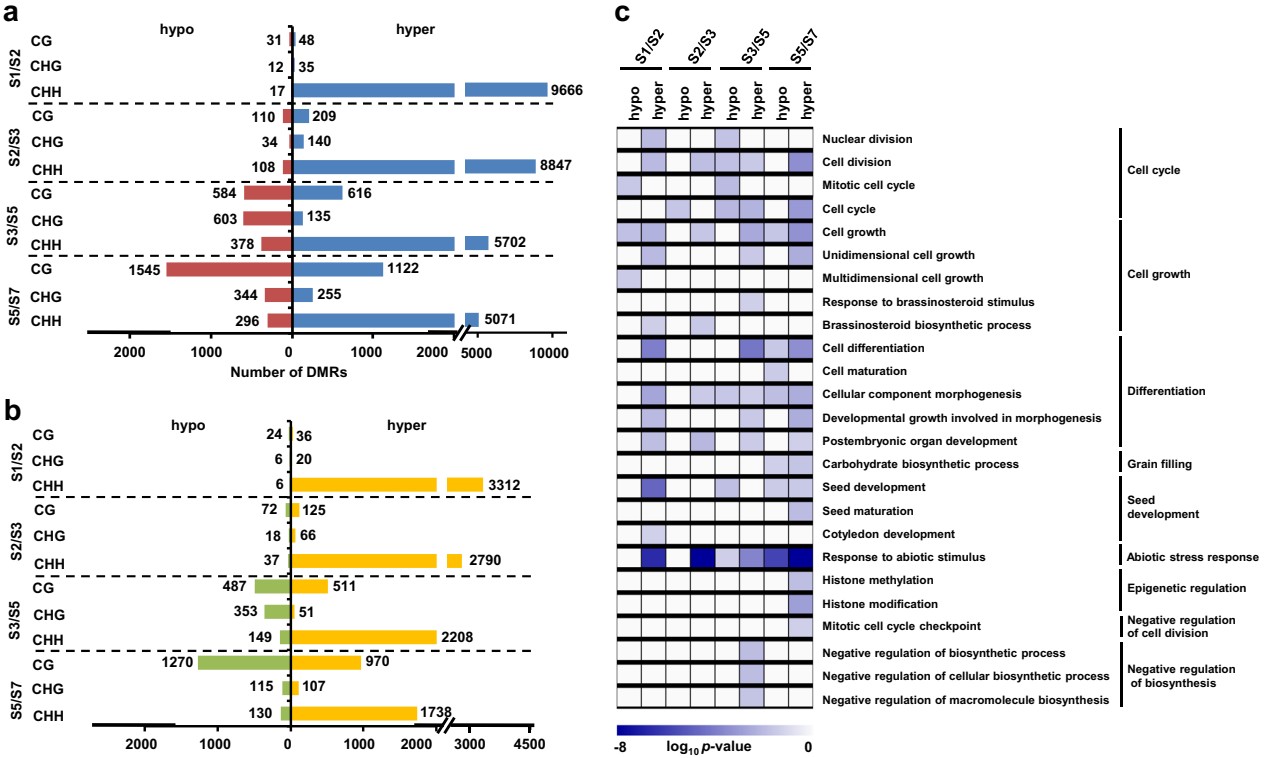

**Fig. 2 Differential methylation during seed development. a, b** Number of differentially (hypo and hyper) methylated regions (DMRs) (**a**) and number of DMR-associated genes (**b**) in CG, CHG, and CHH contexts between successive stages of seed development are shown in bar graphs. **c** Gene ontology (GO) analysis of DMR-associated genes during seed development. Enriched GO (biological process) terms in hypo and hypermethylated sets of genes during successive stages of seed development are shown via heatmap. Scale represents log₁₀ *p*-value of enriched GO terms. The associated biological processes are given on the right side.

hypomethylated genes during successive stages of seed development revealed enrichment of biological process terms, including seed development, cell cycle/cell division, cell growth, differentiation, grain filling processes, epigenetic regulation and abiotic stress response (Fig. 2c).

To examine the influence of differential methylation on differential gene expression, we analyzed expression profiles of DMR-associated genes between successive stages of seed development using RNA-seq data[3]. A total of 382, 311, 872, and 1284 DMR-associated genes exhibited differential expression during S1/S2, S2/S3, S3/S5, and S5/S7 stage transitions, respectively (Fig. 3a). We determined fraction of genes showing differential (hypo/hyper) methylation in different sequence contexts located in different genic regions and their direction of differential (up/down) expression during successive stages of seed development. In general, no consistent relationship between differential methylation in different sequence contexts and/or genic/flanking regions, and direction of differential gene expression was observed (Fig. 3b, c; Supplementary Dataset 3). However, differential methylation in specific sequence context(s) was found related with direction of differential gene expression during specific stage transitions. For example, higher (69.2%) fraction of genes with CG context hypermethylation in their gene body during S2/S3 transition showed higher expression at S3 stage (Fig. 3b). Similarly, hypomethylation in CG (76.9%) and CHG (80%) contexts in upstream region during S3/S5 transition was found correlated with higher gene expression at S5 stage (Fig. 3b). However, the number of genes showing such relationships represented only a minor fraction of the total DMR-associated differentially expressed genes (DEGs). Although majority of

DEGs were found associated with CHH context differential methylation in different regions at all the stage transitions, no obvious trend of correlation between the direction of differential methylation and differential gene expression was observed.

Further, we analyzed influence of differential methylation on differential gene expression for sets of genes involved in important biological processes during seed development. We selected four sets of genes, including those involved in cell cycle, differentiation, grain filling and desiccation processes, based on their associated GO terms (Supplementary Fig. 6). Highest fraction (33.5–56.4%) of DMR-associated genes during S5/S7 transition belonging to these biological processes represented DMR-DEGs (Supplementary Fig. 6a; Supplementary Dataset 4). An inverse relation between the direction of differential methylation and differential gene expression was observed in some instances among these sets. For example, most of CHH context hypermethylated genes involved in cell cycle exhibited downregulation during S1/S2 transition (Supplementary Fig. 6b–d). Hypomethylation in CHG context was correlated with higher expression of genes involved in cell cycle and differentiation processes at S5 stage during S3/S5 transition (Supplementary Fig. 6b–d). Interestingly, desiccation response related genes with CG context hypermethylation within gene body primarily showed higher expression at S7 stage as compared to S5 stage. These results suggest influence of DNA methylation dynamics on transcript abundance of important genes relevant to the seed development processes.

**Reprogramming of DNA methylation in TEs.** Several evidences have shown that DNA methylation mediated TE silencing is

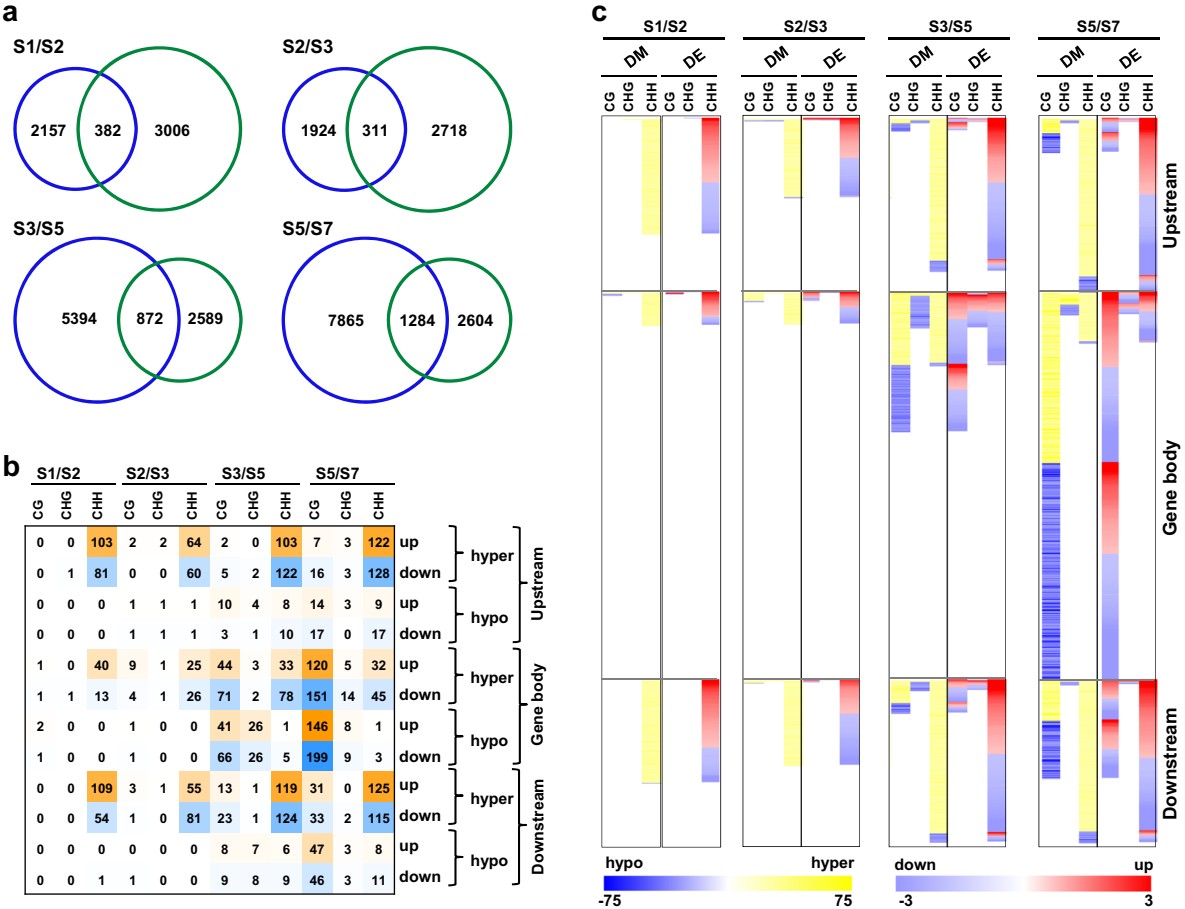

**Fig. 3 Differential methylation and differential gene expression during seed development. a** Number of DMR-associated genes (blue circle) and differentially expressed genes (green circle) between successive stages of seed development are given in the Venn diagrams. **b** Number of differentially expressed (up/down) genes that are differentially methylated (hyper/hypo) in different sequence contexts and genic regions between successive stages of seed development are given. Intensity of mustard and blue colors indicate number of DMR-associated upregulated and downregulated genes, respectively. **c** Differential methylation (DM) and differential expression (DE) of the number of genes given in **b** is shown via heatmaps. Scales at the bottom represent percentage methylation level difference (hypo/hyper) and differential expression (up/down) in $\log_2$ fold-change.

important during seed development[25,26]. We also interrogated the methylation status of TEs in different sequence contexts at different stages of seed development (Fig. 4a; Supplementary Fig. 7; Supplementary Dataset 5a). Methylation level in all the sequence contexts was much higher within TE body regions as compared to their flanking regions (Fig. 4a; Supplementary Fig. 7). However, progressive gain of DNA methylation in TEs was observed throughout seed development in CHH context only (Fig. 4a). Increased methylation level was observed for both class I and class II TEs. Class I TEs representing long terminal repeats (LTRs) exhibited highest methylation level, which is in agreement with a previous report in soybean[38]. However, long interspersed nuclear elements (LINEs), another class I transposons, and DNA transposons (class II), exhibited comparatively lower methylation level (Fig. 4b; Supplementary Dataset 5b). The increase in methylation level was highest during S5/S7 transition as compared to other stage transitions (Fig. 4a, b). All but 0.01% of CHH context DMRs located in class I and class II TEs exhibited hypermethylation during all the seed development stage transitions (Fig. 4c; Supplementary Dataset 5c).

To examine the plausible role of DNA methyltransferases[39] and demethylases in TE methylation, we analyzed their gene expression during seed development. Although three genes encoding DNA methyltransferases (CaDRM1, CaDRM2, and CaCMT1) were highly expressed at S5 and/or S7 stages of seed

development (Fig. 5a), their expression profiles did not explain progressive gain of TE methylation during seed development (Fig. 5a). Gene expression of most of demethylases was found to be highest at S5 stage (Fig. 5a), which also did not explain progressive gain in CHH context methylation.

Next, we performed small RNA sequencing from the same five stages of seed development and non-redundant sets of 21-nucleotide (nt) and 24-nt small RNAs were selected for further analyses (Supplementary Table 2). The density of 21-nt and 24-nt small RNAs was found to be significantly high in the hypermethylated TEs in CHH context as compared to all TEs during all the stage transitions (Fig. 5b, c; Supplementary Fig. 8a, b; Supplementary Dataset 6a), suggesting that small RNAs can play a role in methylation of TEs possibly via RdDM-dependent pathway during seed development. The density of 24-nt small RNAs was much higher than 21-nt small RNAs during all the stage transitions (Fig. 5b, c; Supplementary Fig. 8a, b). Although methylation level was lower, progressive gain of methylation in CHH context in the set of TEs not associated with small RNAs was also observed throughout seed development (Fig. 5d; Supplementary Fig. 8c; Supplementary Dataset 6b). These results indicate that RdDM-independent pathway may complement RdDM-dependent pathway to some extent in TE methylation during seed development as suggested earlier too[19].

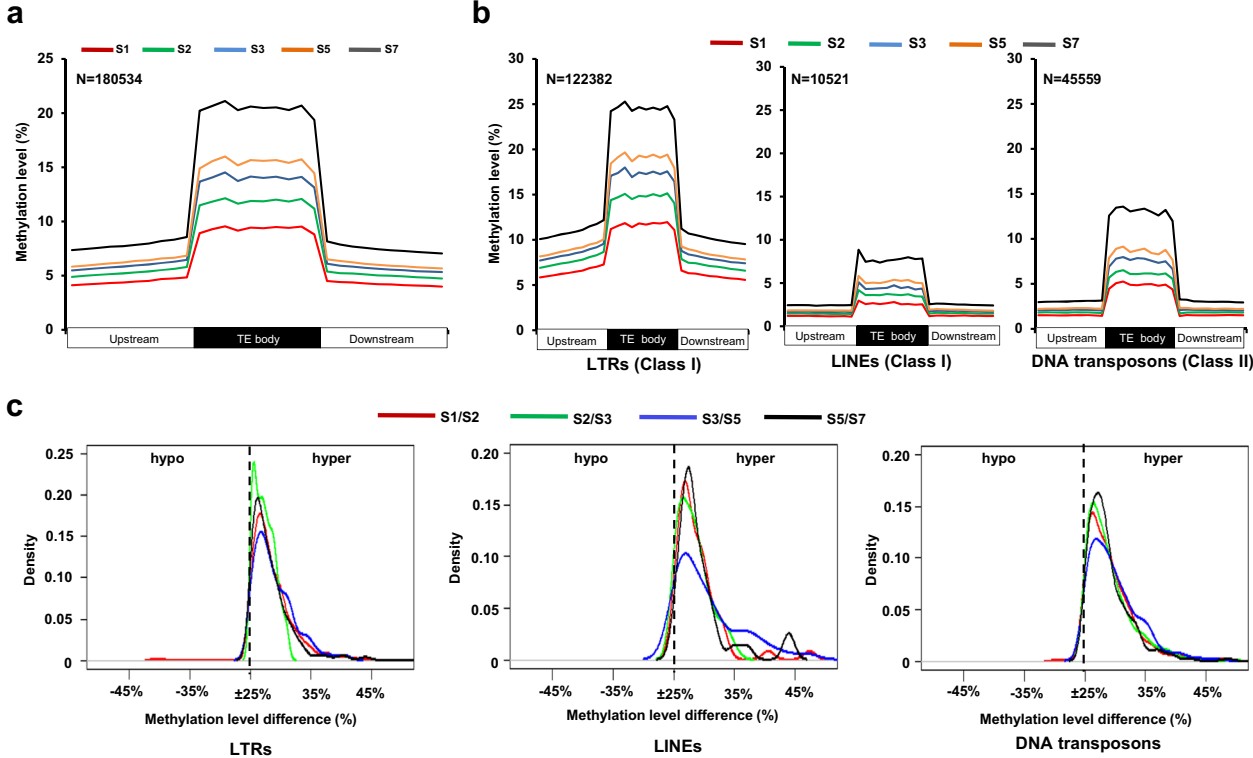

**Fig. 4 Methylation in transposable elements (TEs) during seed development. a** Methylation level within TEs and their 2 kb flanking (upstream and downstream) regions in CHH context is shown at different stages of seed development. Methylation level in CG and CHG contexts is shown in Supplementary Fig. 7. **b** Methylation level in CHH context within TE body and their flanking regions for class I (LTRs and LINEs) and class II (DNA transposons) TEs is shown. Each region was divided into 10 bins of equal size and normalized methylation level in each bin is shown in the line graphs in **a** and **b**. **c** Methylation level difference in the CHH context DMRs located in class I and class II TEs between successive stages of seed development is shown via kernel density plots. The DMRs (hypo/hyper) with significant methylation difference (≥25% methylation level difference with ≤0.01 q-value) were analyzed.

**TEs regulate differential gene expression**. Earlier reports have demonstrated that methylation status of TEs influence expression of proximal genes[40,41]. We interrogated the influence of methylation of TEs in different sequence contexts on expression level of their proximal protein-coding genes and/or genes containing TEs within their body region (intragenic TEs) (Fig. 6a–c). The expression level of genes associated with methylated intragenic TEs in CHG and CHH contexts was much lower than the genes associated with non-methylated intragenic TEs (Fig. 6b, c), which is in contrast to the genes associated with intragenic methylated TEs in CG context (Fig. 6a). In general, expression level of genes associated with methylated TEs in all sequence contexts in their flanking regions were lower than the genes not associated with methylated TEs. These results suggest an important role of TE methylation in influencing expression of proximal and/or overlapping genes.

Next, we examined frequency of TEs in the sets of genes associated and/or not associated with DMRs in different sequence contexts between successive stages of seed development. Interestingly, DMR-associated genes in all sequence contexts harbored significantly higher frequency of TEs within their gene body as compared to genes not associated with DMRs (Fig. 6d–f; Supplementary Dataset 7). In addition, the frequency of TEs within gene body was significantly higher as compared to the flanking regions in CHG and CHH context DMR-associated genes (Fig. 6d–f; Supplementary Fig. 9a). Furthermore, DMR-associated DEGs harbored significantly higher frequency of intragenic TEs in CHH context as compared to DEGs not associated with DMRs (Fig. 6g–i; Supplementary Dataset 7). The frequency of intragenic TEs was also much higher than the

flanking regions in CHH context DMR-associated genes (Fig. 6i; Supplementary Fig. 9b). A few examples of the genes showing differential methylation of intragenic TEs in CHH context and differential gene expression during successive stages of seed development are shown in Supplementary Fig. 10. These observations suggest that differential methylation of intragenic TEs in CHH context contribute to the differential gene expression in large part throughout seed development.

**Differential DNA methylation between small-seeded and large-seeded chickpea**. Based on transcriptome analysis, extended period of cell division and higher level of endoreduplication during late-embryogenesis (S3) and mid-maturation (S5) stages of seed development, respectively, have been suggested to determine seed size/weight in chickpea[3]. To examine the role of DNA methylation in determining seed size/weight in chickpea, we sequenced DNA methylomes of these two stages of seed development in a small-seeded cultivar, Himchana 1 (100 seed weight of 13.15 ± 0.15 g) (Fig. 7a). A total of 109–122 million high-quality read pairs were generated for each sample in Himchana 1 (Supplementary Table 3). About 59–69 million read pairs mapped uniquely, which covered 86–87% of the chickpea genome (Supplementary Table 3). Like JGK 3, higher percentage of methylcytosines were detected in CG and CHG contexts in Himchana 1 too (Fig. 7b). The fraction of methylcytosines in CG and CHG contexts was higher at S3 stage as compared to S5 stage. However, fraction of methylcytosines in CHH context was similar at both stages (Fig. 7b). Average methylation levels were much higher in CG (92.47–92.58%) and CHG (84.59–86.91%) contexts as compared to CHH (41.81–46.62%) context (Fig. 7c;

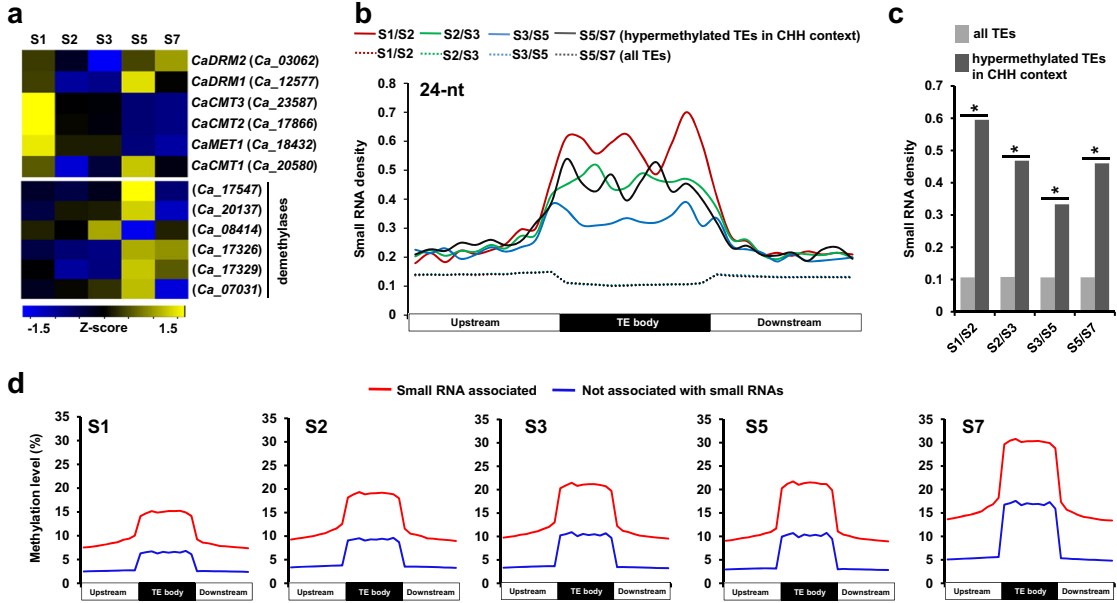

**Fig. 5 Expression profiles of genes encoding DNA methyltransferases and demethylases, and analysis of small RNAs during seed development.**
**a** Heatmap showing expression profiles of genes encoding DNA methyltransferases and demethylases during seed development. Scale represents z-score based on FPKM values. **b** Density of 24-nucleotide (nt) small RNAs in hypermethylated TEs in CHH context and all TEs between successive stages of seed development are shown. Each region was divided into 10 bins of equal size and normalized small RNA density (per 100 bp) in each bin is shown. **c** Density of 24-nt small RNAs within TE body of hypermethylated TEs in CHH context and all TEs is shown. Asterisks denote p-value < 2.2e−16 as determined by Fisher's exact test. **d** Methylation level in CHH context within TE body and 2 kb flanking regions for the TEs associated with 24-nt small RNAs and not associated with small RNAs at different stages of seed development is shown. Each region was divided into 10 bins of equal size and normalized methylation level in each bin is shown.

Supplementary Fig. 11a). No significant methylation level difference between forward and reverse strands was observed (Supplementary Fig. 11b). Like JGK 3, methylation in CHG and CHH contexts within gene body and flanking regions were found to be antagonistically correlated with gene expression levels at both the stages of seed development in Himchana 1 too (Supplementary Fig. 12). Methylation in CG context within the flanking regions of genes also showed antagonistic correlation with gene expression. However, methylation in CG context within gene body showed positive correlation with gene expression. The genes expressed at moderate and high levels exhibited higher methylation level in CG context within their gene body (Supplementary Fig. 12).

Next, we investigated differences in methylation profiles between Himchana 1 (small-seeded) and JGK 3 (large-seeded) cultivars at S3 and S5 stages of seed development. Interestingly, percentage of methylcytosines in Himchana 1 cultivar (56.4% in CG, 41.6% in CHG and 7.8% in CHH contexts) was higher than JGK 3 (50.7% in CG, 39.1% in CHG and 7.2% in CHH contexts) in all the sequence contexts at S3 stage (Fig. 7b). However, percentage of CHG and CHH context methylcytosines was higher in JGK 3 at S5 stage (Fig. 7b; Supplementary Dataset 8). Generally, methylation levels were marginally higher in JGK 3 as compared to Himchana 1 in all the sequence contexts (Fig. 7c).

Further, we identified DMRs in JGK 3 as compared to Himchana 1 (JGK 3/Himchana 1) at S3 and S5 stages of seed development (Fig. 7d). A higher number of DMR-associated genes was detected in CG context at both stages of seed development (Fig. 7e; Supplementary Dataset 9). In total, 9490 hyper and 9724 hypomethylated DMRs representing 6157 hyper and 6008 hypomethylated genes in CG context at S3 stage were detected. Similarly, a total of 2138 hyper and 2017 hypomethylated DMRs representing 1716 hyper and 1619 hypomethylated genes in CG context at S5 stage were detected (Fig. 7d, e;

Supplementary Dataset 9). The number of hypermethylated DMRs and DMR-associated genes in CHH and CHG contexts were higher than hypomethylated ones at both stages of seed development (Fig. 7d, e). Methylation level difference analysis also showed a larger fraction of hypermethylated DMRs in CHG and CHH contexts in JGK 3 cultivar (Supplementary Fig. 13). GO analysis of hyper and/or hypomethylated genes in JGK3 showed enrichment of cell cycle, cell growth, grain filling and seed development related biological process terms, suggesting possible role of DNA methylation in determining seed size/weight in chickpea (Fig. 7f).

To understand the role of DNA methyltransferases and demethylases in determining differential methylation between the two cultivars, we analyzed their differential gene expression between the two chickpea cultivars at S3 and S5 stages of seed development. The transcript levels of CaMET1, CaCMT2, and CaCMT3 was higher at S3 stage, whereas CaDRM1 and CaDRM2 exhibited higher transcripts levels at S5 stage in JGK 3 cultivar as compared to Himchana 1 (Supplementary Fig. 14). Majority of the genes encoding demethylases were expressed at high level at S5 stage in JGK 3 cultivar. However, these results did not explain differential methylation between JGK 3 and Himchana 1 (Supplementary Fig. 14).

**DNA methylation in seed size/weight determination**. To understand the role of DNA methylation in influencing expression of genes involved in seed size/weight determination, we identified DMR-associated DEGs between the cultivars. A total of 1254 and 487 genes at S3 and S5 stages, respectively, were identified as DMR-associated DEGs between the cultivars (Fig. 8a). Among them, genes with CG context differential (hyper/hypo) methylation within gene body were more abundant than those showing differential methylation in other sequence contexts (Fig. 8b–d). We did not observe consistent pattern

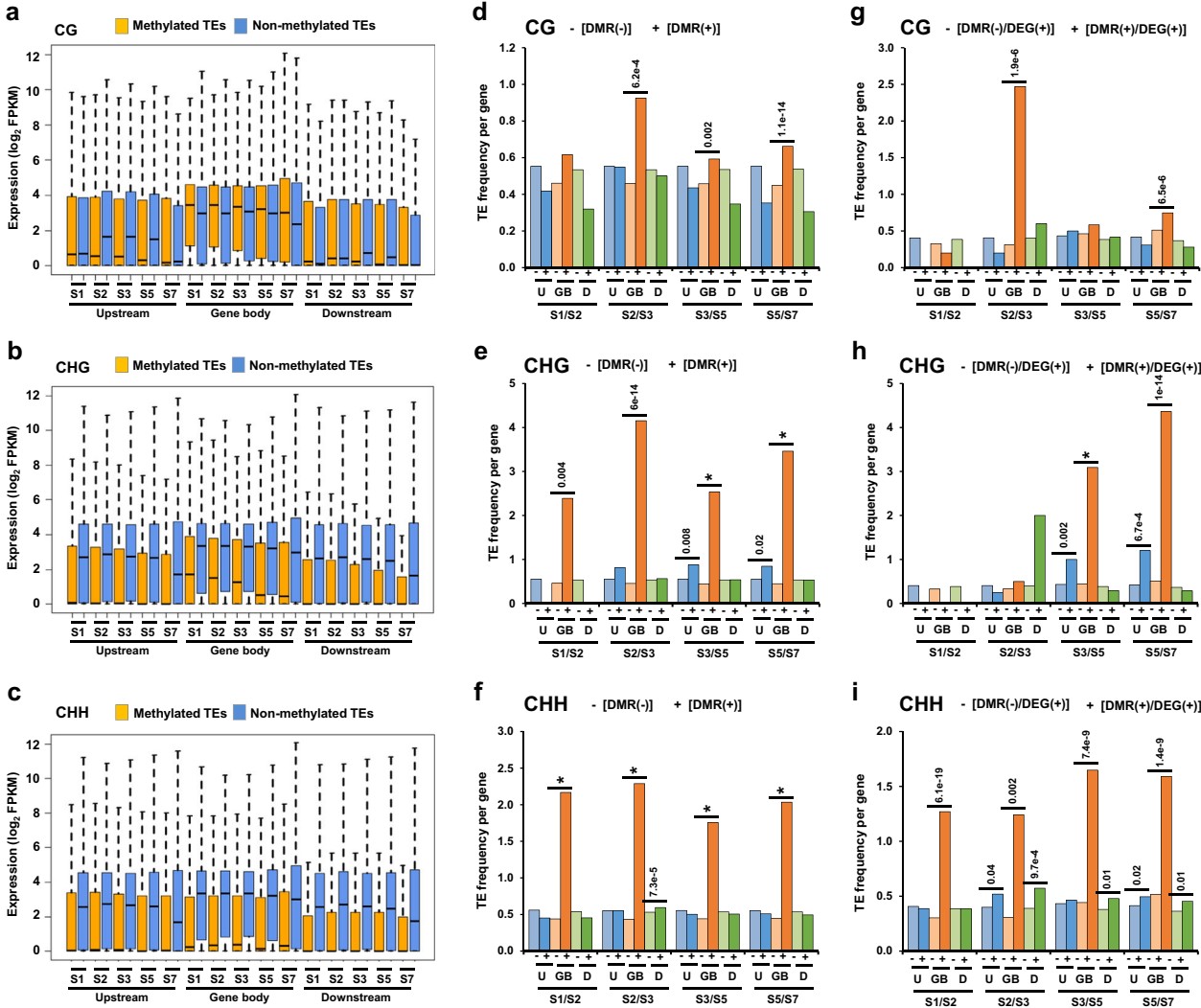

**Fig. 6 Influence of transposable elements (TEs) on differential gene expression during seed development. a–c** Boxplots showing expression level of genes harboring methylated and non-methylated TEs within body and flanking regions in CG (**a**), CHG (**b**), and CHH (**c**) contexts at different stages of seed development. **d–f** Frequency of TEs within gene body and flanking regions of protein-coding genes that are DMR-associated [DMR(+)] and not associated with DMRs [DMR(−)] in CG (**d**), CHG (**e**), and CHH (**f**) contexts is shown in bar graphs. **g–i** Frequency of TEs within gene body and flanking regions of differentially expressed genes that are DMR-associated [DMR(+)/DEG(+)] and not associated with DMRs [DMR(−)/DEG(+)] in CG (**g**), CHG (**h**), and CHH (**i**) contexts is shown in bar graphs. Significance of difference (p-value) was determined by Wilcoxon signed rank test and is indicated above the horizontal bars. Asterisks denote p-value < 2.2e−16. U upstream, GB gene body, D downstream.

between the direction of differential methylation in different sequence contexts located in different genic regions and differential gene expression at both the stages of seed development (Fig. 8b–d; Supplementary Dataset 10). Next, we investigated influence of TEs on differential expression of their associated and proximal genes. A higher frequency of intragenic TEs in DMR-associated DEGs as compared to DEGs not associated with DMRs was observed in all the sequence contexts at both the stages (Supplementary Fig. 15).

Further, we analyzed trend between direction of differential methylation and differential expression for the sets of genes involved in cell cycle, cell growth and grain filling processes, and genes located within known quantitative trait loci (QTLs) associated with seed size/weight in chickpea[42–47] (Supplementary Fig. 16; Supplementary Dataset 11). A higher fraction of DMR-associated DEGs was identified at S3 stage as compared to S5 stage for all the four sets (Supplementary Fig. 16a). However, no clear pattern between the direction of differential methylation and differential gene expression was detected for any of these sets

of genes (Supplementary Fig. 16). In addition, higher frequency of TEs detected in the DMR-associated DEGs in comparison with DEGs not associated with DMRs for the four sets of genes was not found to be significant (Supplementary Fig. 17). These results indicate a limited role of TEs in mediating differential methylation and differential expression of these sets of genes that determine seed size/weight.

Increasing evidences suggest that CG context hypermethylation within gene body results in higher gene expression levels[36,37]. We also observed differential methylation in CG context within gene body mostly represented among the DMR-associated DEGs at both the stages of seed development. In total, 53 genes involved in cell cycle, cell growth, grain filling and genes located within known QTLs-associated with seed size/weight, showed hyper-methylation in CG context within gene body and higher expression in JGK 3 cultivar at S3 and/or S5 seed development stages (Fig. 8e; Supplementary Fig. 18). Among them, 12, 8, 4, and 21 genes involved in cell cycle, cell growth, grain filling and QTL associated genes, respectively, were represented at S3 stage

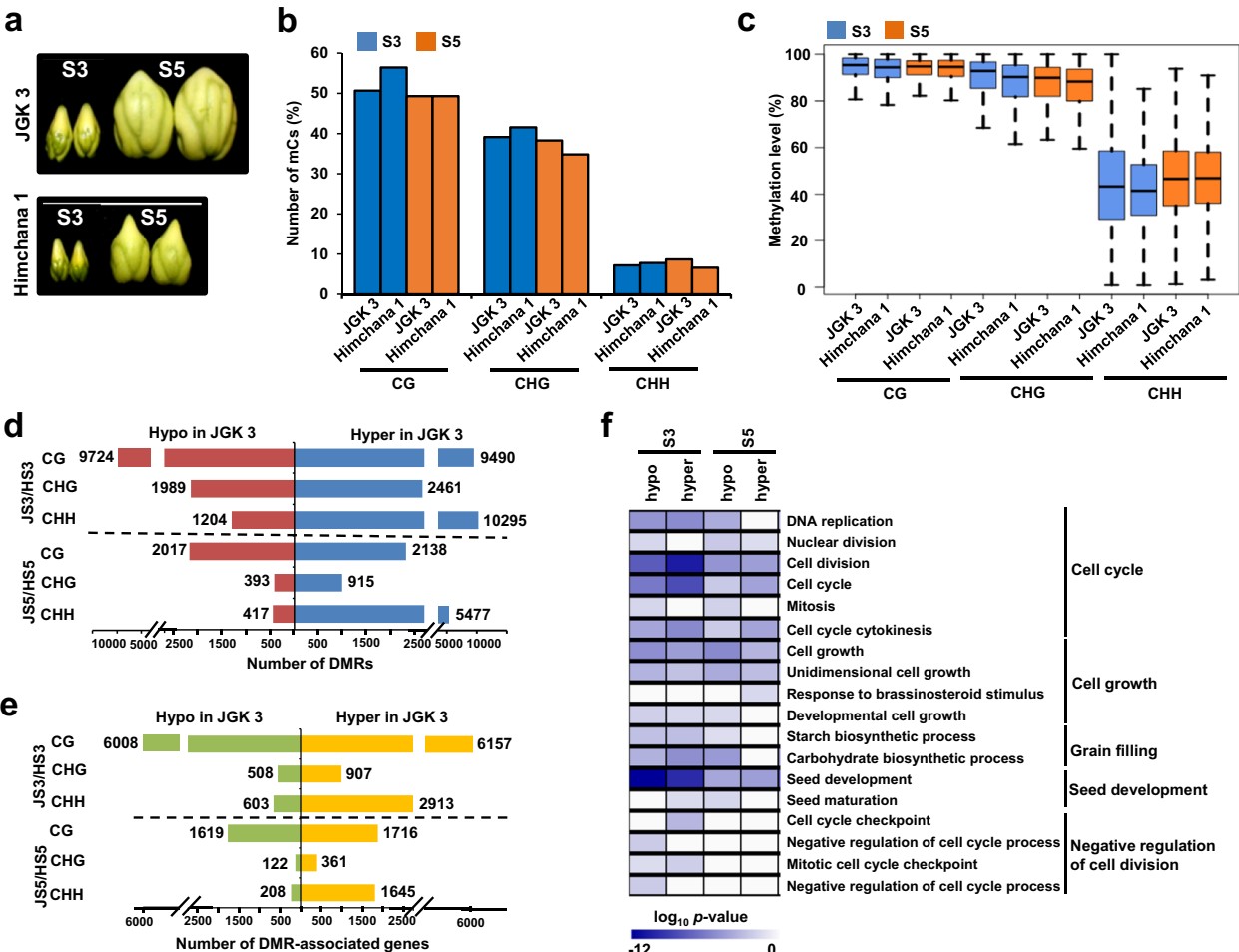

**Fig. 7 Differential methylation between JGK 3 (large-seeded) and Himchana 1 (small-seeded) chickpea cultivars. a** Two stages of seed development, late-embryogenesis (S3) and mid-maturation (S5) in JGK 3 and Himchana 1 cultivars[3] used for comparative DNA methylation analysis. **b** Percentage of methylcytosines (mCs) in different sequence contexts at S3 and S5 stages of seed development in JGK3 and Himchana 1 cultivars is shown in bar graph. **c** Methylation level at individual mC in different sequence contexts at both stages of seed development in the two cultivars is shown via boxplot. **d, e** Number of differentially methylated regions (DMRs) (**d**) and number of DMR-associated genes (**e**) identified between JGK 3 and Himchana 1 cultivars at S3 (JS3/HS3) and S5 (JS5/HS5) stages of seed development are given in different sequence contexts via bar graphs. **f** Gene ontology (GO) analysis of DMR-associated genes at S3 and S5 stages between JGK 3 and Himchana 1. GO (biological process) terms enriched in sets of hyper and hypo methylated genes in JGK 3 as compared to Himchana 1 at S3 and S5 stages of seed development is shown via heatmap. Scale represents log₁₀ *p*-value of enriched GO terms. The associated biological processes are given on the right side.

(Fig. 8e). Similarly, 5 and 4 genes involved in cell cycle and genes associated with QTLs, respectively, were represented at S5 stage (Fig. 8e). One QTL-associated gene (*Ca_09238*) was common at both S3 and S5 stages.

Among the genes showing hypermethylation in CG context within gene body and higher expression at S3 and/or S5 stages of seed development in JGK 3, five genes encoded transcription factors, including type-B response regulator (ARR-B, *Ca_14780*), auxin response factor (ARF, *Ca_10748*), dof zinc finger domain protein (*Ca_09238*), C3HC4-type zinc finger protein (*Ca_09815*) and homeobox domain protein (HB, *Ca_04491*) (Fig. 8e; Supplementary Fig. 18). These transcription factors may be involved in seed development and seed size/weight determination, as they can further regulate their target genes. In addition, genes involved in cell cycle regulation, including cell division cyclase 6 (CDC6, *Ca_15618*) and minichromosomes (*Ca_09549*, *Ca_15408*, and *Ca_12014*) were represented. Two genes (*Ca_06937* and *Ca_23631*) representing members of early auxin-responsive GH3 gene family showed hypermethylation and higher expression in JGK 3, suggesting important role of

auxin signaling in determining seed size/weight via DNA methylation. At least 24 genes, including those encoding for dof zinc finger transcription factor (*Ca_09238*), ARF (*Ca_10748*), RNA recognition motif containing protein (*Ca_09981*) and F-box protein (*Ca_01760*) located within known QTLs associated with seed size/weight, exhibited hypermethylation in CG context within gene body and higher expression at S3 and/or S5 stages in JGK 3 (Fig. 8e).

**Discussion**

Here, we sought to understand the role of DNA methylation during seed development and seed size/weight determination in chickpea via bisulphite sequencing. A progressive gain of DNA methylation in CHH context was observed during successive stages of seed development. Concomitantly, increased methylation during seed development was majorly found in TEs. Previous reports have demonstrated that TEs are silenced via methylation in CHH context in gametes and embryo[17,21–24]. In addition, progressive gain of methylation in TEs throughout seed development in Arabidopsis and soybean have also been reported[25,26].

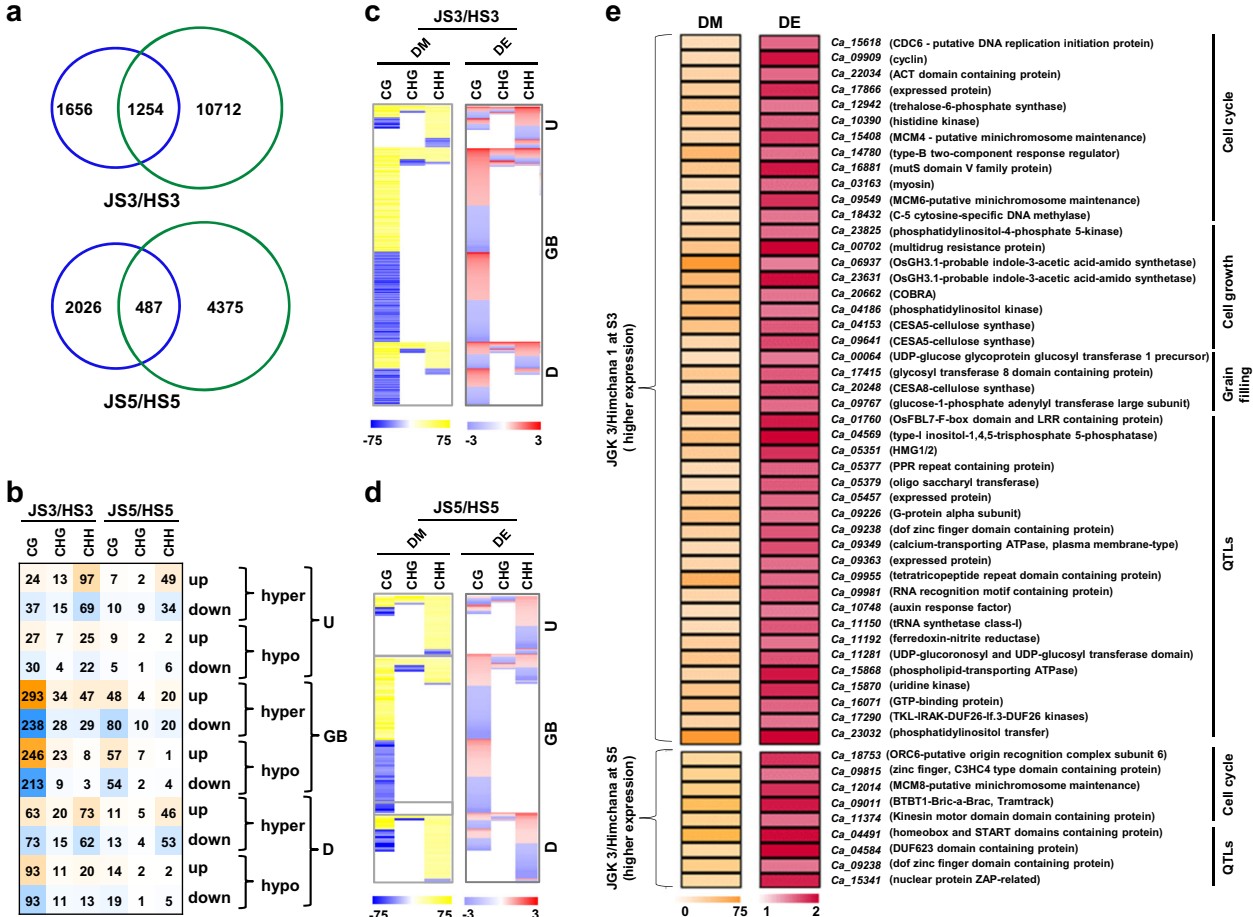

**Fig. 8 Differential methylation and differential gene expression between JGK 3 and Himchana 1 chickpea cultivars. a** Number of DMR-associated (blue circle) and differentially expressed genes (green circle) between Himchana 1 and JGK 3 at S3 and S5 stages of seed development are given in Venn diagrams. **b** Number of differentially (hyper/hypo) methylated and differentially (up/down) expressed genes in different sequence contexts and genic regions between Himchana 1 and JGK 3 at S3 and S5 stages of seed development are shown. Intensity of mustard and blue colors indicate number of DMR-associated upregulated and downregulated genes, respectively. **c, d** Differential methylation (DM) and differential expression (DE) of the genes between JGK 3 and Himchana 1 at S3 (**c**) and S5 (**d**) stages of seed development given in **b** are shown via heatmaps. Scales at the bottom represent percentage of methylation level difference and differential expression in log$_2$ fold-change. **e** Heatmap showing hypermethylation in CG context within gene body in JGK 3 (as compared to Himchana 1) associated with higher expression at S3 and/or S5 stages of seed development for sets of genes involved in cell cycle, cell growth, grain filling processes and genes located within known QTLs-associated with seed size/weight in chickpea. Scales at the bottom represent percentage of methylation level difference and differential expression in log$_2$ fold-change. U upstream, GB gene body, D downstream.

These reports along with our results suggest that TE methylation during seed development is common phenomenon in plants.

Previous studies have shown that global demethylation in CG context due to loss of MET1 function result in abnormal embryo development in Arabidopsis and severe necrotic lesions in rice seedlings[28,29]. However, global demethylation in CHG and CHH contexts did not exhibit abnormalities in the plants[26]. These results suggested that DNA methylation in CG context may play crucial role during seed development. However, no correlation between differential methylation in any sequence context and differential gene expression during seed development was observed in Arabidopsis and soybean[25,26]. We also did not observe consistent pattern of relationship between direction of differential methylation in different sequence contexts and differential gene expression during seed development in chickpea. However, differential methylation in the sets of genes involved in cell cycle, differentiation and desiccation in specific sequence context(s), and/or genic region(s) was found to be correlated with differential gene expression during successive stages of seed development. This suggested that DNA

methylation in specific set of genes can play a role during seed development.

Previous studies showed that both 21-nt and 24-nt small RNAs guide methylation of TEs in gametes and embryo[17,21–24]. We found higher density of both 21-nt and 24-nt small RNAs in CHH context hypermethylated TEs, suggesting role of small RNAs in TE methylation during seed development. However, 24-nt small RNAs seem to play a major role in TE repression due to their higher frequency. Higher methylation level in CHH context along with its progressive gain in TEs associated with small RNAs during successive stages suggested role of RdDM-dependent pathway in seed development. However, the role of RdDM-independent pathway can also not be ruled out due to progressive gain of methylation in the TEs not associated with small RNAs too, which is in accordance with a previous study[19]. It will be interesting to study that which cell type(s)/tissue(s) reinforce methylation in TEs during seed development. Although, endosperm is completely resorbed before reaching S5 stage in chickpea, possibility of TE methylation at the later stages of seed development via small RNAs originated from endosperm during

embryogenesis and storing them for later use cannot be ruled out completely.

Previous studies have shown that imprinted genes are regulated via differential methylation within TEs located proximal to protein-coding genes in allele-specific manner[48,49]. Our results showed that frequency of intragenic TEs was much higher than frequency of TEs located in proximal regions in DMR-associated DEGs between successive stages of seed development. This suggest that intragenic TEs can influence expression of their associated genes to a larger extent. A previous study also suggested possible role of intragenic TEs in regulating expression of their associated genes in Arabidopsis[50].

The role of DNA methylation in seed size/weight determination is largely unknown. We compared DNA methylome profiles between JGK 3 (large-seeded) and Himchana 1 (small-seeded) chickpea cultivars at late-embryogenesis (S3) and mid-maturation (S5) stages of seed development, which were found to be most important in determining seed size/weight[3]. Differential methylation was mostly found in CG context between the two cultivars, which may be due to selection/diversification of DNA methylation marks mostly in CG context in cultivar-specific manner. A large fraction of DEGs were associated with DMRs in CG context within gene body. Increasing evidences showed that hypermethylation in CG context within gene body may increase gene expression levels[36,37]. Further, it has been demonstrated that intragenic DNA methylation can prevent spurious transcription initiation by RNA polymerase II[51]. These results suggest that CG context hypermethylation in gene body in JGK 3 cultivar may govern transcription of genes involved in seed size/weight determination. We revealed a total of 53 candidate genes involved in cell cycle/cell division, cell growth, grain filling processes, and/or genes located within known QTLs associated with seed size/weight showing hypermethylation and higher expression at S3 and/or S5 stages of seed development in JGK 3. Genes involved in cell cycle/division, carbohydrate metabolism, transcription factors and signal transduction pathways were found to be hypermethylated in CG context within gene body and expressed at higher levels in JGK 3. Our data suggest that a few transcription factors and genes encoding signal transduction components may be governed by DNA methylation. Moreover, genes involved in cycle/division and starch biosynthesis may further be regulated by transcription factors and/or signaling component(s) in association with DNA methylation. In addition, identification of differentially methylated and differentially expressed genes located within known QTLs highlighted regulation of seed size/weight determination in chickpea cultivar(s) by both genetic and epigenetic mechanisms to certain extent.

In conclusion, we revealed DNA methylation dynamics during seed development in a large-seeded chickpea cultivar and compared with a small-seeded cultivar. Small RNA-mediated DNA methylation in TEs was found to be associated with seed development. The plausible role of DNA methylation in intragenic TEs associated with differential gene expression during seed development has been unraveled. Substantial difference in CG context DNA methylation between small and large-seeded chickpea cultivars was detected. Hypermethylation in CG context within gene body was found to be associated with higher expression of candidate genes involved in seed size/weight determination in the large-seeded cultivar. Overall, we provide insights into DNA methylation mediated regulation of seed development and revealed candidate genes that may be involved in seed size/weight determination in chickpea.

## Methods

**Plant material and genomic DNA isolation**. Seeds undergoing five landmark stages of development, including early-embryogenesis (S1), mid-embryogenesis (S2), late-embryogenesis (S3), mid-maturation (S5), and late-maturation (S7), of a large-seeded cultivar (JGK 3) of chickpea were collected from the field-grown plants as described previously[3]. Similarly, seeds representing S3 and S5 stages were collected from a small-seeded chickpea cultivar (Himchana 1). At least 30–40 seeds for S1, S2, and S3 stages, and 18–20 seeds for S5 and S7 stages were collected for each biological replicate. The seeds were briefly frozen in liquid nitrogen and stored at −80 °C for later use. The seed samples for transcriptome analysis in previous study[3] and seed samples used in this study were collected together at the same time. The quantity and integrity of the genomic DNA extracted using DNeasy kit (Qiagen, GmbH, Hilden, Germany) were verified using Qubit Fluorimeter (Life Technologies) and agarose gel electrophoresis, respectively.

**Whole genome bisulphite sequencing**. Sample preparation and bisulphite sequencing were carried out as described previously[35,52]. Genomic DNA from each tissue sample was fragmented to a mean size of 200–300 bp via sonication (Covaris, Massachusetts, USA). TrueSeq-methylated adapters were ligated to the ends of fragmented DNA and treated with sodium bisulphite as per manufacturer's recommendations (DNA Methylation-Gold™ kit, Zymo Research Corporation, CA, USA). Sequencing was performed using Illumina HiSeq platform in paired-end mode to generate 100-nt long reads with >30× sequencing depth of chickpea genome for each sample. At least two biological replicates were sequenced for each stage of seed development analyzed in both the cultivars (Illumina, San Diego, USA).

**Read alignment and identification of methylcytosines**. The raw reads were processed to remove reads with adapter sequences and low-quality bases using NGSQC Toolkit (v2.3)[53] using default parameters. The high-quality filtered reads were mapped to kabuli chickpea genome (v1)[54] using Bismark (v0.14.3)[55] with default parameters and only the reads mapped at unique position were retained. All high-quality reads were aligned to the chloroplast genome (naturally unmethylated) of chickpea too to estimate the efficiency of bisulphite conversion and error-rate. The non-conversion of chloroplast genome Cs to Ts was considered as a measure of error rate. We calculated $p$-value for each cytosine covered by sequencing in the chickpea genome using binomial test, and criteria of $p$-value ≤0.0001 and sequencing depth of ≥5 reads were used for identification of true *methylcytosines* as described in previous studies[35,52]. Methylation level at each *methylcytosine* site was determined by percentage of reads giving methylation call (C) to all the reads aligned (C and T) at the same site[35,52]. Methylation level in genes/TEs and 2 kb flanking regions was determined using perl scripts. Each gene/TE body and its flanking regions were partitioned into ten bins of equal size and average methylation level in each bin was determined by normalizing to the number of cytosines present in the respective bin. The TEs harboring at least one *methylcytosine* per 100 bp were considered as methylated TEs and the TEs that do not contain *methylcytosines* were considered as nonmethylated.

**Gene sets and TEs**. A total of 28,269 genes and 180,535 TEs predicted in the chickpea genome[54] were used for different analyses unless otherwise mentioned. Gene body refers to the genomic sequence from start to stop coordinates of each gene given in the gff file. The 2 kb flanking sequences from *transcription start site* and *transcription termination site* of each gene represented the upstream and downstream regions, respectively. TE body included genomic sequence from start to stop coordinates of each TE. The sets of genes involved in different biological processes, including cell cycle, differentiation, cell growth, grain filling and desiccation, were identified based on their GO term assignment. The genes encoding methyltransferases and demethylases were identified in kabuli chickpea based on the previous study[39] and gene annotation search, respectively. Class I (LTRs and LINEs) and class II (DNA transposons) TEs were designated based on their available annotation[54]. QTL-associated genes were identified based on the previous studies[42–47] as described earlier[3].

**Identification of DMRs and DMR-associated genes/TEs**. DMRs between successive stages of seed development in JGK 3 and between the two cultivars (JGK 3/ Himchana 1) at S3 and S5 stages of seed development were identified using 100 bp window size and step size of 50 bp in the chickpea genome based on sliding window approach using methylkit package (v0.2.5) in R. The bins with at least three cytosines and covered with sequencing depth of ≥5 reads that showed ≥25% methylation level difference with ≤0.01 $q$-value (corrected $p$-value using Sliding Linear Model), were considered as differentially methylated bins as described previously[35,52]. Consecutive differentially methylated bins located within 50 bp were merged to identify DMRs in their respective sequence contexts. The DMRs showing at least 25% methylation level difference with ≤0.01 $q$-value during transition to successive stages of seed development and/or between the two cultivars at any stage(s), were used for drawing the kernel density plots. The genes with overlapping DMRs (determined based on their genomic co-ordinates using Bedtools) within 2 kb flanking or body regions were identified as DMR-associated (hypomethylated or hypermethylated) genes for each comparison. If a DMR was located across upstream region and gene body or gene body and downstream region, it was considered in both categories. Likewise, frequency of TEs per gene was determined based on their overlap with different genic regions. The difference

in frequency of TEs between different sets of genes or genomic regions with *p*-value ≤0.05 (determined using Wilcoxon signed rank test) was considered to be statistically significant. The comparative distribution of *methylcytosines* in different sequence contexts, DMRs and expression profile between the two stages of seed development in JGK 3 or between JGK 3 and Himchana 1 along with gene and/or TE annotation, in example set of genes was visualized via Integrative Genomics Viewer (IGV, v2.4.14).

**Gene ontology enrichment analysis**. Enrichment of GO (biological process) terms was carried out using BiNGO tool at Cytoscape (v3.7). Genes associated with hyper and/or hypomethylated DMRs during successive stage transitions in JGK 3, and between Himchana 1 and JGK 3 at S3 and S5 stages of seed development, were analyzed. A cut-off of ≤0.05 *p*-value significance was used to identify enriched GO terms.

**Integration of DNA methylation and gene expression analysis**. RNA-seq data representing the same stages of seed development (S1, S2, S3, S5, and S7 in JGK 3, and S3 and S5 in Himchana 1) in chickpea cultivars from our previous study[3] were used for integration with DNA methylation. At all the stages, genes were categorized into 10 sets (deciles) based on their FPKM expression values, wherein 1st and 10th deciles represented the sets of non-expressed genes and genes expressed at highest levels, respectively. Methylation level within genes/TEs and 2 kb flanking regions from *transcription start site* (upstream) and *transcription termination site* (downstream) were estimated for these sets of genes. Differential gene expression analysis between successive stages of seed development (S1/S2, S2/S3, S3/S5, and S5/S7) within JGK3, and between JGK 3 and Himchana 1 at S3 and S5 stages was performed using Cuffdiff (v2.0.2) as described previously[3]. The genes showing at least two-fold change with *q*-value ≤0.05 were defined as differentially expressed genes. The direction of differential methylation (hypo/hyper) in different sequence contexts and genic regions of DMR-associated genes, and their differential expression (up/down) between successive stages within JGK 3 cultivar, and between JGK 3 and Himchana 1 cultivars, was analyzed.

**Small RNA sequencing and data analysis**. Total RNA was isolated from seeds at S1, S2, S3, S5, and S7 stages in JGK 3 using TRI reagent as described in our previous study[3]. The quantity and quality of the RNA were verified using Bioanalyzer (Agilent technologies, CA, USA). High-quality RNA was used to prepare small RNA sequencing library as per manufacturer's recommendations (Illumina technologies). Each sample was sequenced in single-end sequencing mode to obtain 50-nt long reads. Raw data files were processed to remove adapter sequences using Cutadapt[56]. High-quality reads representing exactly the same sequences were collapsed into unique read(s). The unique reads were mapped on chickpea genome using Bowtie (v1.1.2) with no mismatch allowed. The reads mapped to non-coding sequences, structural RNAs (rRNA, tRNAs and snoRNAs) and chickpea chloroplast genome were removed as described in a previous study[57]. The remaining 21-nt and 24-nt long reads representing small RNAs were selected for further analysis. Density of small RNAs was determined based on overlap of their middle base with TEs and/or their 2 kb flanking regions. Each TE and its flanking regions were partitioned into ten bins of equal size and small RNA density in each bin (per 100 bp) was analyzed. The difference in small RNA density between different sets of TEs with *p*-value ≤0.05 (determined using Fisher's exact test) was considered to be statistically significant.

**Statistics and reproducibity**. The details about experimental design and statistics used in different data analyses performed in this study are given in the respective sections of results and methods. For bisulphite sequencing, we used at least two independent biological replicates of different seed development stages of JGK 3 and Himchana 1. For each cytosine of the genome covered by bisulphite sequencing, *p*-value of methylation level was determined using binomial test in each sample. The statistical significance of differential methylation for each bin of the genome (100 bp) was determined by calculation of *p*-value using logistic regression test followed by *q*-value using Sliding Linear Model as implemented in Methylkit. Cuffdiff software was used to determine the significance (*p*-value by *t*-test followed by *q*-value by Banjamini-Hochberg method) of differential expression between the two stages. The assessment of enrichment of GO terms was done via hypergeometric test using BiNGO plugin of Cytoscape. The statistical significance of difference in frequency of TEs between different sets of genes/genomic regions was determined using Wilcoxon signed rank test. The statistical significance of difference in small RNA density was determined using Fisher's exact test.

**Reporting summary**. Further information on research design is available in the Nature Research Reporting Summary linked to this article.

## Data availability
Bisulphite-sequencing data for large-seeded and small-seeded cultivars have been deposited at NCBI's Gene Expression Omnibus (GEO) and is accessible via series accession numbers GSE131665 and GSE131669, respectively. Small RNA sequencing data have also been deposited at GEO and is accessible via series accession number

GSE131424. RNA sequencing data used in this study is available via series accession numbers GSE79719 and GSE79720 at GEO.

## Code availability
Details of publicly available software used in the study are given in the "Methods". No custom code or mathematical algorithm that is deemed central to the conclusions was used.

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

## Acknowledgements

This work was financially supported by the Department of Biotechnology, Government of India, New Delhi (BT/AB/NIPGR/SEED BIOLOGY/2012). M.J. acknowledges Tata Innovation Fellowship from the Department of Biotechnology, Government of India, New Delhi. We are thankful to Dr. V. Singh for help with collection of tissue samples. M.S.R. thankfully acknowledges Science and Engineering Research Board, New Delhi for National Postdoctoral Fellowship. K.G. acknowledges research fellowship from Shiv Nadar University.

## Author contributions

M.J. and R.G. designed and supervised the experiments and data analyses. M.S.R., K.G., N.K.K., R.G. and M.J. performed and/or analyzed the data. M.S.R. and M.J. wrote the manuscript. All authors discussed the results, revised the draft manuscript, and read and approved the final manuscript.

## Competing interests

The authors declare no competing interests.
