## [Peer Review File · Communications Biology]

Reviewers' comments:

Reviewer #1 (Remarks to the Author):

The manuscript by Rajkumar and colleagues describes a study in which they have mapped the dynamics of gene expression, DNA methylation and small RNAs across chickpea seed development. They also assess differences in each of these between two chickpea cultivars, one large and one small seeded, attempting to correlate the differences with the seed phenotypes. The data are interesting and are well analysed and described, though there are some limitations that I will specify below. The manuscript makes a useful contribution to our understanding of molecular events during seed development and does so in a less well studied species.

There are a number of findings described in the manuscript. The data demonstrate well that differential methylation occurs reasonably extensively during chickpea seed development. The most strong changes are a progressive gain of CHH methylation and loss of CG methylation. Many genes differentially expressed between developmental stages are differentially methylated, though there is not a clear trend in the context or direction of methylation/expression correlation (ie. hypo/hyper methylation v. Up/down regulation). The authors highlight changes in CG methylation that correlate with differential expression (Fig. 3b), but they do not discuss a reasonably large number of genes that have differential CHH methylation in promoters; this result should be discussed also. Overall, many differentially expressed genes are differentially methylated. In an interesting piece of analysis the authors find that genes both differentially methylated and differentially expressed between developmental stages tend to be closely associated with greater numbers of transposons, either proximal to or within the genes. Comparison of methylomes between large and small seeded cultivars did not identify a specific association of some region of methylation with seed size. I am unsurprised by this - you would need an experiment designed differently, for example an epiRIL population, to find a causal locus. Nonetheless, the comparative analysis of the methylomes between cultivars is useful to researchers in the field.

The Discussion reaches too far and requires substantial revision. The authors return repeatedly to the idea that they have defined a role for DNA methylation and epigenetics in chickpea seed development, and that DNA methylation regulates chickpea seed development. The final paragraph of the Discussion is the clearest example, but they do so throughout. Regulate has a very specific definition, and none of the experiments described in this manuscript have examined regulation. What the authors have done is very nicely chart the dynamics of methylation and identify interesting correlations between methylation and expression. The Discussion should be revised to highlight this instead; determining regulatory function would be a different study involving mutants and the like.

Subject to appropriate revision of the Discussion, plus addressing some detailed points below, I am in favour of publication.

Detailed point to address

- The Methods are not detailed enough to recreate the analyses, particularly regarding the bioinformatics. Which software packages were used with which parameters and what statistical thresholds? This is crucial.
- In40 - should specify in Arabidopsis
- I do not understand Supp Fig 4 - GO analyses. How do the terms in table relate to rows in figure? Needs to be redrawn and explained more clearly.
- Supp Fig 5 shows "correlations" between mC and DE sets of genes without any statistics. Fig needs

to include numbers of genes per group and also measures of statistical significance for the correlations.

- Fig.5b, association between sRNAs and 24mers. The dotted lines appear to be missing for all but one of the transitions. Necessary to understand distinction between all TEs and hypermethylated only.

Again, section asserts "significantly" without a statistical test. Revise.

-ln224 - RdDM independent... not associated with sRNAs. This needs to be described with consideration of the possibility it is a technical (mapping) artefact, because sRNAs are short and are to map. Again, methods not sufficiently detailed - what number of mismatches was permitted when mapping sRNA? Often it is 0, but that may create false negatives.

-ln229 - proximal TEs and methylation. What level of methylation was defined as non-methylated (state in methods).

- Fig6b - there are open bars behind the main bar. What do these mean?

- ln249-253. Differentially methylated genes that are DE are associated with transposons. Again, "significantly" without a measure of significance.

- Fig7d - should not be labelled hypo/hyper, because hyper in which cultivar? Label hyper in A / hyper in B or similar. The scaling on the axes is also confusing - is 500 before or after the break point on 7d and what value does the breakpoint restart from?

- ln286-294 suffers same problem - hyper/hypo in which cultivar?

-Supp Fig 10. I don't understand - number of terms does not correspond to number of rows or columns on plot. Please clarify.

- Similarly hard to interpret Supp Fig. 11. Hypo/hyper in which cultivar?

- Fig 8c/d. Description on line 336 does not agree with the legend. Description says up in JGK3, but figure indicates both up and down. Which is it?

- ln353-4 - genes don't encode indole acetic acid, it's not a protein.

Reviewer #2 (Remarks to the Author):

The authors have generated an impressive dataset of matching RNA-seq and whole genome methylC-seq data from chickpea seed development. The analyses, mostly, follow a logical theme to show the impact of methylation reprogramming during seed development in a large seeded cultivar and also via comparison to a small seeded cultivar. Especially laudable is the effort to use beautiful data display to convey the major conclusions that 1) Differential methylation occurs for both genes and TEs during seed development, 2) There are more TEs within or near genes with changed expression, and 3) Large seeded cultivar has higher CG methylation than smaller seeded cultivar for a set of important genes that play a role in seed size/weight determination. I have specific concerns with certain methodology and choice of control which if and when addressed will improve the quality and readability of this manuscript.

Major comments:

1. For Figure 1C, and subsequent metaplots, additional information is required. A) What is the length of gene body that all genes are scaled to? B) What exactly is the gene body – transcriptional unit, or protein coding region or the whole region including introns? C) How many genes are part of this metaplot? – similarly for TEs later in Figure 4, how many TEs are part of each figure panel. D) The metaplot analysis hides the fact if the effect is due to a small change in methylation for a large group of genes or a large change in methylation for a small group of genes. The authors should consider including either a confidence interval around the average or have a supporting heatmap showing the percentage of genes that undergo the observed change in average methylation.

2. The methylation density is calculated as number of mCs in 100 bp intervals. This method is inherently flawed since different regions of the genome may have different base composition (one region may have fewer cytosines). The authors must fix this issue and calculate either percentage

methylation by normalizing to the number of Cs at the very least. For better approach the depth of each cytosine should also be factored in (Schultz et. al. Trends in Genetics, 2013).

3. For kernel density plots shown in Figure 2C, the center of the x-axis should be 0, with -25% and +25% on either side of the plots. I do not understand the reason for shrinking the plot. The authors should either expand on the reason for this representation or present a figure with a linear x-axis and no broken axis.

4. To investigate the hypermethylation of TEs, the authors analyzed the expression of genes encoding DNA methyltransferases. The authors should also analyze the possibility of loss in expression of DNA demethylases which can also explain increase in methylation genome-wide.

5. Line 229-253 (Section on TEs regulate differential gene expression) – This section tries to demonstrate that TEs regulate differential gene expression, but the controls can be better analyzed. The experimental set is to investigate the expression of genes with methylated intragenic TEs or with neighboring TEs. The control sets can either be 1- unmethylated intragenic TEs (and neighboring TEs) or 2- absence of intragenic TEs (or neighboring TEs) or both. The authors chose set 1 as control set for intragenic TEs and set 2 as control for neighboring TEs. Either the authors should stay consistent across and choose the same set as control or use both control sets.

6. Line 252-253: Simply showing that there are more intragenic TEs in DMR-associated DEGs, does not conclusive show that intragenic TEs play an important role in gene regulation. The authors should show that methylation change in intragenic TEs for these sets is correlated with gene expression change or refrain from over interpreting the data.

7. For many of the figures, the number of genes in the data is not shown. N should be mentioned to make it clear average of how many genes are being compared.

8. An opportunity is missed by the authors. They show that TEs are hypermethylated during seed development but fail to show if specifically, the subset of intragenic TEs or gene-neighboring TEs undergo the same hyper methylation, and whether this hypermethylation cause change in expression across seed development. Isn't it the central story of the manuscript that epigenetic reprogramming affects gene expression? but in spite of spending considerable space and time analyzing TEs, the authors fail to directly show that link through TEs. Additionally, are unmethylated TEs getting methylated or previously methylated TEs getting hypermethylated is also an important question that can be easily answered and help link methylation of TEs and genes with proper controls. This is a suggestion which can improve the manuscript but it is up to the authors if they want to investigate this.

9. Line 337-341: The authors try to conclude that genes from specific sets (cell cycle, cell growth etc) are represented in DMR-associated DEGs but a corresponding number for how many DEGs not associated with DMRs for this set is not provided, which is required to conclusively show that methylation changes specifically alters expression level.

10. The authors must show at least a few single gene examples, for which there is conclusive evidence of increase in methylation and corresponding decrease of expression or vice-versa and if possible the involvement of intragenic or neighboring TEs. All the analyses are done on a genome-wide scale which is the strength of the paper, but a few 'good examples' in the supplementary figures will increase the confidence of highly processed bioinformatics data figures.

Minor comments:

1. The authors have submitted the sequences to GEO but these accessions are not accessible yet. I understand they can be kept private until publication but the authors can request a reviewer's token specifically for the reviewers to review the submission. I urge the authors to include reviewer's token for current and future manuscript submissions.

2. Line 204: The authors should refrain from conclusion regarding TE length without doing the corresponding TE length analysis.

3. The title mentions epigenetic reprogramming but realistically the authors have only investigated DNA methylation and I strongly suggest to change the title to include 'Methylation' reprogramming.

Reviewer #3 (Remarks to the Author):

The authors studied methylation changes during seed development in chickpea large-seeded cultivar (JGK 3) and small-seeded cultivar (Himchana 1). A comparison of DNA methylomes between JGK 3 and a small-seeded cultivar (Himchana 1) revealed that 53 candidate genes involved in seed size/weight determination exhibiting hypermethylation in CG context within gene body and higher expression in JGK 3. The authors concluded that DNA methylation mediated regulation of seed development and seed size/weight determination in chickpea. This is an interesting conclusion. However, there are several concerns.

The first major concern is that the current analysis result cannot fully support the statement. The major conclusion of this manuscript is, as shown in the abstract, "Overall, this study provides insights into DNA methylation mediated regulation of seed development and seed size/weight determination in chickpea." However, the data cannot support the statement as indicated in the comment (2) and (3). The overstatement includes the following statements.

(1) Line 232:

The expression levels of genes associated with methylated intragenic TEs was significantly lower than the genes associated with non-methylated intragenic TEs except at S2 stage (Fig. 6a).

=>

(i) The authors did not examine if these TEs are located in intron or exon.

(ii) The authors did not go deeper to examine if these genes are TE-like genes. Genome annotation may annotate TE remnants as genes, because the flanking repeat sequences experienced heavy mutation leading to not being able to be predicted as TEs and the middle TE proteins are still predicted by gene prediction program. As a result, some genes are "TE genes" derived from TEs.

(iii) The authors did not go deeper to examine if these genes are pseudogenes with specific criteria like ENCODE project with specific criteris to define a pseudo gene.

(iv) When examining the expression profile in Fig 6(a), 75% of the genes with methylated TEs in gene bodies are silent except S2. These group of genes could be a mixtures of real functional genes, pseudogenes and TE-like genes.

(v) Without deeper analysis indicated above, the current analysis cannot support the statement that "The expression levels of genes associated with methylated intragenic TEs was significantly lower than the genes associated with non-methylated intragenic TEs except at S2 stage".

(2) Line 329

...suggesting important role of intragenic TEs in seed size/weight determination in these sets of genes

=>

It is overstatement. The result just shows that, using grain filling DEGs as example, grain filling DEGs can be divided into two groups, (i) some are not differentially methylated, and, (ii) some are differentially methylated. That's all. The authors may examine that in the grain filling DGEs, the gene number of DMRs-associated ones are statistically significantly higher than the ones with no DMR association.

(3) Line 358

These results suggest that DNA methylation play an important role in regulating candidate genes

involved in seed size/weight determination in chickpea cultivar(s).

=>

- (i) It is known that gene with medium high expression levels are CG-gene body methylated (Nat Genet. 2007;39:61-9).
- (ii) In bigger seeds, relative genes sets (grain filling etc) are higher expressed than in small seeds and they are DEGs.
- (iii) Thus, it is not surprised that these grain filling genes have gene body methylation in big seeds.
- (iv) There are many possibilities there are CG gene body methylation in genes with medium high RNA levels and the role is still enigmatic (Curr Opin Plant Biol. 2017;36:103-110). Also one possibility is that the CG gene body methylation occurs in these highly expressed genes relative to big seeds phenotypes in order to prevent spurious transcription initiation (Nature. 2017;543:72-77).
- (v) Together, the authors have no sufficient data to fully support the statement that "DNA methylation play an important role in regulating candidate genes involved in seed size/weight determination".

Second, there is no definition of many terms used in this manuscript leading to not possible to judge if the analysis result can fully support the statement. This includes the following statements.

(4) Line 143:

Methylation dynamics and its impact on differential gene expression during seed development
=> What are definition of promoter regions and downstream regions in this analysis in terms of distance to transcription start site and termination site? Besides, if one DMR is located across promoter and gene body, then the this is counted as promoter or gene body? Same as downstream and gene body. Please clarify it.

(5) Line 288

The number of hypomethylated DMRs were marginally higher than hypermethylated DMRs in JGK 3...
=> No definition the DMR is derived from which cultivar minus which one. Then not possible to judge if the data can support the statement.

(6) Line 308:

Further, GO (biological process) analysis of hyper and/or hypomethylated genes showed enrichment of cell cycle,

=> There is no definition of hyper and/or hypomethylated genes in this manuscript. Is it DMR-associated genes?

The third major concern is derived from no detailed information leading to not possible to judge if the data can fully support the statement. This includes the following statements.

(7) Line 179:

Further, we analyzed the influence of differential methylation on differential gene expression for sets of genes involved in important biological processes during seed development. We selected four sets of genes, including those involved in cell cycle, differentiation, grain filling and desiccation processes based on their associated GO terms (Supplementary Fig. 5).

=> MAIN CONCERNS: the statement is the role of DNA methylation in regulating important sets of genes involved in important biological processes during seed development. However, it is not clear what genes are exactly tested in this analysis.

- (i) What are genes used in the analysis to get the result of Supplemental Fig 5? The authors should show the gene list.
- (ii) What's the criteria to choose genes in this analysis? It is based on the reference papers? If yes,

please list the reference papers.

(8) Line 240 to line 248

MAJOR CONCERNS: No sufficient information and analysis to judge if the analysis result can support the statement or not.

DMR associated genes harboured significantly higher frequency of TEs within their gene body during all the stage transitions (Fig. 6b)...

=>

(i) The DMRs here should be divided into CG, CHG and CHH contexts to reveal more information. If there are many CHG and CHH DMRs here, then it may suggest these DMRs are derived from TEs. If it is true that there are many CHG and CHH DMRs, then it is not surprising that DMR associated genes have higher number of TEs as compared non-DMR associated genes.

(ii) In Fig 6(b) and SFig 7 (b) and (c), the authors did statistic test here. The authors should show the original data of the what four numbers here were used for doing Fischer exact test as supplemental table. Thus, the readers have better idea how the authors did analysis to get the result.

The forth major concern is that the statement made by analysis which has no appropriate control as below.

(9) Line 251:

Interestingly, frequency of intragenic TEs was significantly higher in the DMR-associated DEGs in all the four sets of genes analyzed too (Supplementary Fig. 7)

=>

The authors need a control for this analysis: doing the same analysis in DMR associated genes in promoter and downstream regions. Without the control, the conclusion (intragenic TEs play important role in regulating gene regulation during seed development) is suspicious.

The following is the "novelty" issue.

In abstract:

(10) Higher frequency of small RNAs in hypermethylated TEs in successive stages suggested role of RNA-dependent DNA methylation (RdDM) pathway.

=> Annu Rev Plant Biol. 2015;66:243-67.

(11) Progressive gain of DNA methylation in CHH context in transposable elements (TEs) was observed during seed development.

=> Proc Natl Acad Sci U S A. 2017 Nov 7;114(45):E9730-E9739.

(12) In The 2nd result section: Influence of DNA methylation on gene expression during seed development

=> It's been known in Nat Genet. 2007 Jan;39(1):61-9.

(13) The forth result section: Epigenetic reprogramming of TEs during seed development

The conclusion is "These results suggest that RdDM-independent pathway also complement RdDM dependent pathway to some extent in TE methylation during seed development".

=> It's been known in Cell. 2013 Mar 28;153(1):193-205.

The following is the "statistic test" issue.

(14) Line 157:

The methylation level differences were more significant in CG and CHG contexts during S3/S5 and

S5/S7 transitions (Fig. 2c).

=> What is statistic test and cutoff p value in this analysis?

(15) Line 172:

Significant correlation was observed in specific sequence context(s) during specific stage transitions

=> What statistic test was done here?

The following is minor concerns in introduction.

(16) Line 40

Epigenetic modifications control reorganization of chromatin structures to determine euchromatic or heterochromatic regions driven by internal and/or environmental cues

=> need to add reference

The followings are minor concerns in results.

(17) Line 96: "We analyzed early-embryogenesis (S1), mid-embryogenesis (S2), late-embryogenesis (S3), mid-maturation (S5) and late maturation (S7) stages"

=> Add seed photo as Fig 1 in Plant J. 91(6), 1088–110 (2017) to help the readers have better idea what they are.

(18) Line 103:

"only 0.006% read pairs mapped to the 104 chloroplast genome, which confirmed high efficiency of bisulphite conversion in our 105 experiments (Supplementary Table 1)."

=> Is it the un-conversion rate? The way to calculate un-conversion rate is to compare the reference genome sequence vs BS-seq sequence at base resolution, rather than "reads map the chloroplast genome". Please clarify it.

(19) Line 120:

...context was observed during seed development (Fig. 1c). The gain of methylation was much higher at the S7 stage especially in the distal flanking regions (>500 bp from genic ends).

=> No scale on the x-axis leading to not being possible to determine the location of 500 bp. Please add the 500 bp position on the x-axis of the figure.

(20) Line 170:

At global level, no consistent correlation pattern between differential methylation and differential gene expression at all the successive stages of seed development, was observed...

=> There are so many numbers in the fig 3b and 3c. It is very confusing what the authors exactly mean "At global level"?

(21) Line 177:

However, the number of genes showing correlation represented only a minor fraction of the total DMR-associated DEGs...

=>

(i) Does this "correlation" means positive correlation only? Or negative correlation? Or together?

(ii) What does this mean exactly? Does this mean "significant correlation was observed in specific sequence context(s) during specific stage transitions" is only a minor fraction? If using statistic test to

examine the fraction of genes are "significant correlation was observed in specific sequence context(s) during specific stage transitions" to conclude that "the number of genes showing correlation represented only a minor fraction of the total DMR-associated DEGs", then the authors should make a summary table to show the fraction of genes which pass the statistic test cutoff value.

(22) Line 286

Further, we identified DMR-associated DEGs between the two cultivars (JGK3/Himchana1) at S3 and S5 stages of seed development

=>

(i) It is DMR-associated "genes" or "DEGs"? Should be "genes", right? Because in line 315, you indicated "we identified DMR-associated DEGs between the cultivars" and talked about DMR-associated DEG for the entire paragraph after line 315. But in Fig 7d for this paragraph, you indicated it is "genes". This error occurs in the critical sentence makes the readers very confusing which cripples the readers in terms of understating the content.

(ii) it is very confusing what is the definition of the DMRs-associated genes? DMRs in promoter? Gene body? Downstream regions?

(iii) Not sure the logic flow for the entire paragraph. This sentence saying "DMR-associated DEGs" were identified, but the following entire paragraph is nothing relative to DEGs, but DMRs and DMR-associated genes only.

(23) Line 287

higher fraction of DMRs were detected in CG context at both the stages of seed development

=> Higher than what? What to compare?

(24) Line 290

9734 hypomethylated DMRs

=> 9734 here, but 9724 in the fig 7a. Please clarify it.

The followings are minor concerns in figures and tables.

(25) Fig. 1b

=> Not clear how to get the boxplot. Is it the single C site? Or the sliding window with specific size?

(26) Fig. 3b, c:

=> (i) Between Fig 3a and 3b, the number of genes cannot match with each other. For example, in fig 3a, 382 genes in S1/S2 are DMR-associated genes and DEGs. However, in fig 3b, the sum of the gene numbers in S1/S2 is higher than 382.

(ii) Besides, what is the color intensity of green and red boxes?

(27) Fig. 4b:

=> Fig 4b: the promoter of TE is in the repeat ends, not outside the TE. Thus, the "promoter" on the fig 4b should be corrected.

(28) Fig. 6a:

It is suspicious that that the expression levels of genes with methylated TEs insertion in gene bodies

are as similar to genes with non-methylated TEs insertion in gene bodies. The authors may check it again.

(29) Supplementary Table 2:

=> The small RNA number in Stab 2 is the normalized result or not? Please clarify it.

(30) Supplementary Fig. 2

=> Not clear how to get the boxplot. Is it the single C site? Or the sliding window with specific size?

(31) Supplementary Fig. 3

=> Please add label of S2 -S7 on the fig.

(32) Supplementary Fig. 4:

=> The heat map does not match the corresponding GO terms on the right part of the heatmap.

(33) Supplementary Fig. 10

=> Fig on the left does not fit the table on the right.

(34) Supplementary Fig. 11

=> The heat map figure is twisted.

In summary, data in this this manuscript cannot support some statements, especially the major conclusion, "Overall, this study provides insights into DNA methylation mediated regulation of seed development and seed size/weight determination in chickpea." Furthermore, missing detailed information and well-defined terms used in this manuscript results in not possible to judge if the analysis results can support the statements. On the other hand, some statements are not novel. Finally this manuscript has many editing errors. I would suggest the authors to re-think about the analysis results and make the statement carefully.

Reviewers' comments:

Reviewer #1 (Remarks to the Author):

The manuscript by Rajkumar and colleagues describes a study in which they have mapped the dynamics of gene expression, DNA methylation and small RNAs across chickpea seed development. They also assess differences in each of these between two chickpea cultivars, one large and one small seeded, attempting to correlate the differences with the seed phenotypes. The data are interesting and are well analysed and described, though there are some limitations that I will specify below. The manuscript makes a useful contribution to our understanding of molecular events during seed development and does so in a less well studied species.

There are a number of findings described in the manuscript. The data demonstrate well that differential methylation occurs reasonably extensively during chickpea seed development. The most strong changes are a progressive gain of CHH methylation and loss of CG methylation. Many genes differentially expressed between developmental stages are differentially methylated, though there is not a clear trend in the context or direction of methylation/expression correlation (ie. hypo/hyper methylation v. Up/down regulation). The authors highlight changes in CG methylation that correlate with differential expression (Fig. 3b), but they do not discuss a reasonably large number of genes that have differential CHH methylation in promoters; this result should be discussed also. Overall, many differentially expressed genes are differentially methylated. In an interesting piece of analysis the authors find that genes both differentially methylated and differentially expressed between developmental stages tend to be closely associated with greater numbers of transposons, either proximal to or within the genes. Comparison of methylomes between large and small seeded cultivars did not identify a specific association of some region of methylation with seed size. I am unsurprised by this - you would need an experiment designed differently, for example an epiRIL population, to find a causal locus. Nonetheless, the comparative analysis of the methylomes between cultivars is useful to researchers in the field.

The Discussion reaches too far and requires substantial revision. The authors return repeatedly to the idea that they have defined a role for DNA methylation and epigenetics in chickpea seed development, and that DNA methylation regulates chickpea seed development. The final paragraph of the Discussion is the clearest example, but they do so throughout. Regulate has a very specific definition, and none of the experiments described in this manuscript have examined regulation. What the authors have done is very nicely chart the dynamics of methylation and identify interesting correlations between methylation and expression. The Discussion should be revised to highlight this instead; determining regulatory function would be a different study involving mutants and the like.

Subject to appropriate revision of the Discussion, plus addressing some detailed points below, I am in favour of publication.

Reply: Many thanks for constructive and encouraging comments. We have now revised the MS including discussion taking into consideration all the comments/suggestions.

Detailed point to address

- The Methods are not detailed enough to recreate the analyses, particularly regarding the bioinformatics. Which software packages were used with which parameters and what statistical thresholds? This is crucial.

Reply: Many thanks for the suggestion. We have now provided more details of the methods used, including software packages with parameters and statistical thresholds in the revised MS.

- ln40 - should specify in Arabidopsis

Reply: We have now specified “in Arabidopsis” as suggested and added relevant references.

- I do not understand Supp Fig 4 - GO analyses. How do the terms in table relate to rows in figure? Needs to be redrawn and explained more clearly.

Reply: This seems to have happened during PDF conversion of the MS. We ensured to upload the correct version this time. Please note that the said Figure has been moved as Fig. 2c.

- Supp Fig 5 shows "correlations" between mC and DE sets of genes without any statistics. Fig needs to include numbers of genes per group and also measures of statistical significance for the correlations.

Reply: We have now added the number of genes shown in the heatmap in Supplementary Fig. 6 (earlier Supplementary Fig. 5). The correlation mentioned in the MS referred to the relationship between the direction of differential methylation (hyper/hypo) and differential gene expression (up/down). We have now moderated the sentences to clarify this.

- Fig.5b, association between sRNAs and 24mers. The dotted lines appear to be missing for all but one of the transitions. Necessary to understand distinction between all TEs and hypermethylated only. Again, section asserts "significantly" without a statistical test. Revise.

Reply: The dotted lines appear to be missing for all but one stage transitions due to essentially similar density of 24-nt small RNAs in all TEs at all the stages. We have now moderated the relevant text and figure legend to clarify distinction between all TEs and hypermethylated TEs in CHH context. Further, we have now shown the statistical significance of the difference in density of small RNAs between the two sets of TEs during all stage transitions (Fig. 5c).

-ln224 - RdDM independent... not associated with sRNAs. This needs to be described with consideration of the possibility it is a technical (mapping) artefact, because sRNAs are short and are to map. Again, methods not sufficiently detailed - what number of mismatches was permitted when mapping sRNA? Often it is 0, but that may create false negatives.

Reply: Many thanks for the suggestion. The details of small RNA data processing and mapping have now been provided in “Methods” section. We used perfect match (allowing no mismatch) as criteria for mapping. Although the possibility of some false negatives due to the mapping criteria cannot be ruled out completely, it seems unlikely to affect the overall results as it remains true for all small RNAs (associated and not associated) at all genomic positions.

-ln229 - proximal TEs and methylation. What level of methylation was defined as non-methylated (state in methods).

Reply: The required details have been added in the “Methods” section now.

- Fig6b - there are open bars behind the main bar. What do these mean?

Reply: We are sorry for the confusion. Actually, they were not the open bars, but the lines joining the two bars to show the statistical significance level of the difference between the two values. We have now revised the figure with straight line(s) showing significance level between the two values.

- ln249-253. Differentially methylated genes that are DE are associated with transposons. Again, "significantly" without a measure of significance.

Reply: Many thanks for the suggestion. We have moderated the sentence now.

- Fig7d - should not be labelled hypo/hyper, because hyper in which cultivar? Label hyper in A / hyper in B or similar. The scaling on the axes is also confusing - is 500 before or after the break point on 7d and what value does the breakpoint restart from?

Reply: We are sorry for the confusion. We have revised the figure with labels as hyper/hypo in JGK 3 (as compared to Himchana 1) and the same has been mentioned clearly in the text too. We have corrected the position of break points in Figures 7.

- ln286-294 suffers same problem - hyper/hypo in which cultivar?

Reply: We have revised the text to make it clear.

-Supp Fig 10. I don't understand - number of terms does not correspond to number of rows or columns on plot. Please clarify.

Reply: This seems to have happened during PDF conversion of the MS. We ensured to upload the correct version this time. Please note that the said Figure has been moved as Fig. 7f.

- Similarly hard to interpret Supp Fig. 11. Hypo/hyper in which cultivar?

Reply: Many thanks for the suggestion. We have now clearly mentioned hyper and hypomethylated in JGK 3 (as compared to Himchana 1) in the revised MS.

- Fig 8c/d. Description on line 336 does not agree with the legend. Description says up in JGK3, but figure indicates both up and down. Which is it?

Reply: Many thanks for the comment. It was an inadvertent mistake. The figure reference has been changed to Fig. 8e in the revised MS.

- ln353-4 - genes don't encode indole acetic acid, it's not a protein.

Reply: We have revised the sentence now.

Reviewer #2 (Remarks to the Author):

The authors have generated an impressive dataset of matching RNA-seq and whole genome methylC-seq data from chickpea seed development. The analyses, mostly, follow a logical theme to show the impact of methylation reprogramming during seed development in a large seeded cultivar and also via comparison to a small seeded cultivar. Especially laudable is the effort to use beautiful data display to convey the major conclusions that 1) Differential methylation occurs for both genes and TEs during seed

development, 2) There are more TEs within or near genes with changed expression, and 3) Large seeded cultivar has higher CG methylation than smaller seeded cultivar for a set of important genes that play a role in seed size/weight determination. I have specific concerns with certain methodology and choice of control which if and when addressed will improve the quality and readability of this manuscript.

Reply: Many thanks for a thorough review of the MS and encouraging comments.

Major comments:

1. For Figure 1C, and subsequent metaplots, additional information is required. A) What is the length of gene body that all genes are scaled to? B) What exactly is the gene body – transcriptional unit, or protein coding region or the whole region including introns? C) How many genes are part of this metaplot? – similarly for TEs later in Figure 4, how many TEs are part of each figure panel. D) The metaplot analysis hides the fact if the effect is due to a small change in methylation for a large group of genes or a large change in methylation for a small group of genes. The authors should consider including either a confidence interval around the average or have a supporting heatmap showing the percentage of genes that undergo the observed change in average methylation.

Reply: The suggested details and description have been added in the revised MS at relevant places. For quick reference, (A) Gene/TE body and its flanking regions were partitioned into ten bins of equal size to determine the normalized methylation level and/or small RNA density in each bin. (B) Gene body represented the genomic sequence between start and stop coordinates as per gff file (the region that span exons and introns). (C) Metaplots in Fig. 1d and Figure 4a were generated for all the annotated genes (28269) and TEs (180535) TEs, respectively, in the chickpea genome. The descriptions have been added in the revised MS. (D) We have now shown that average methylation levels in different genic regions and sequence contexts are within the interquartile range (1st and 3rd quartiles) (Supplementary Fig. 4), which reflects that the effect is not due to a small change in methylation for a large group of genes or a large change in methylation for a small group of genes.

2. The methylation density is calculated as number of mCs in 100 bp intervals. This method is inherently flawed since different regions of the genome may have different base composition (one region may have fewer cytosines). The authors must fix this issue and calculate either percentage methylation by normalizing to the number of Cs at the very least. For better approach the depth of each cytosine should also be factored in (Schultz et. al. Trends in Genetics, 2013).

Reply: We have reanalyzed and revised all the metaplots showing methylation level after normalizing to the total number of Cs in each bin, as suggested. The depth of each cytosine has also been factored in, while calculating the methylation level at each cytosine site.

3. For kernel density plots shown in Figure 2C, the center of the x-axis should be 0, with -25% and +25% on either side of the plots. I do not understand the reason for shrinking the plot. The authors should either expand on the reason for this representation or present a figure with a linear x-axis and no broken axis.

Reply: We identified the DMRs with $\geq 25\%$ methylation level difference and q-value cut-off of ≤ 0.01 . Therefore, we presented the methylation level differences of DMRs with $\geq 25\%$ (for hyper) or $\leq -25\%$ (for hypo) and q-value cut-off of ≤ 0.01 in the kernel density plots. The description has been added in the revised MS in the text and legends for clarity. Please note that the said Figure has been moved as Supplementary Fig. 5 in the revised MS.

4. To investigate the hypermethylation of TEs, the authors analyzed the expression of genes encoding

DNA methyltransferases. The authors should also analyze the possibility of loss in expression of DNA demethylases which can also explain increase in methylation genome-wide.

Reply: We analyzed expression profiles of genes encoding DNA demethylases too in the revised MS and presented the heatmaps in Fig. 5a and Supplementary Fig. 14, as suggested. However, their expression profile also did not explain the methylation changes/patterns observed in the study.

5. Line 229-253 (Section on TEs regulate differential gene expression) – This section tries to demonstrate that TEs regulate differential gene expression, but the controls can be better analyzed. The experimental set is to investigate the expression of genes with methylated intragenic TEs or with neighboring TEs. The control sets can either be 1- unmethylated intragenic TEs (and neighboring TEs) or 2- absence of intragenic TEs (or neighboring TEs) or both. The authors chose set 1 as control set for intragenic TEs and set 2 as control for neighboring TEs. Either the authors should stay consistent across and choose the same set as control or use both control sets.

Reply: We have now reanalysed the data with appropriate controls and presented the results along with statistical significance of different comparisons in the revised MS in Fig. 6 and Supplementary Fig. 9, as suggested. The results have been added in the text accordingly.

6. Line 252-253: Simply showing that there are more intragenic TEs in DMR-associated DEGs, does not conclusive show that intragenic TEs play an important role in gene regulation. The authors should show that methylation change in intragenic TEs for these sets is correlated with gene expression change or refrain from over interpreting the data.

Reply: Many thanks for the suggestion. We have now compared the frequency of intragenic TEs with those located in the flanking regions of the four sets of DMR-associated DEGs [DMR(+)/DEG(+)] genes analyzed in revised Supplementary Fig. 9. We have revised the sentence as suggested.

7. For many of the figures, the number of genes in the data is not shown. N should be mentioned to make it clear average of how many genes are being compared.

Reply: Many thanks for the suggestion. We have now added total number of genes/TEs (N) analyzed, as suggested.

8. An opportunity is missed by the authors. They show that TEs are hypermethylated during seed development but fail to show if specifically, the subset of intragenic TEs or gene-neighboring TEs undergo the same hyper methylation, and whether this hypermethylation cause change in expression across seed development. Isn't it the central story of the manuscript that epigenetic reprogramming affects gene expression? but in spite of spending considerable space and time analyzing TEs, the authors fail to directly show that link through TEs. Additionally, are unmethylated TEs getting methylated or previously methylated TEs getting hypermethylated is also an important question that can be easily answered and help link methylation of TEs and genes with proper controls. This is a suggestion which can improve the manuscript but it is up to the authors if they want to investigate this.

Reply: Many thanks for the suggestions. We have now shown that hypermethylation in intragenic TEs contribute to the differential expression in large part during seed development.

9. Line 337-341: The authors try to conclude that genes from specific sets (cell cycle, cell growth etc) are represented in DMR-associated DEGs but a corresponding number for how many DEGs not associated with DMRs for this set is not provided, which is required to conclusively show that methylation changes specifically alters expression level.

Reply: We have now provided the number of DEGs associated and not associated with DMRs for different sets of genes analyzed (Supplementary Fig. 16a).

10. The authors must show at least a few single gene examples, for which there is conclusive evidence of increase in methylation and corresponding decrease of expression or vice-versa and if possible the involvement of intragenic or neighboring TEs. All the analyses are done on a genome-wide scale which is the strength of the paper, but a few 'good examples' in the supplementary figures will increase the confidence of highly processed bioinformatics data figures.

Reply: Many thanks for the suggestion. We have now shown the examples of a few selected genes that show differential methylation and differential expression during seed development and those involved in seed size/weight determination in Supplementary Fig. 10 and 18.

Minor comments:

1. The authors have submitted the sequences to GEO but these accessions are not accessible yet. I understand they can be kept private until publication but the authors can request a reviewer's token specifically for the reviewers to review the submission. I urge the authors to include reviewer's token for current and future manuscript submissions.

Reply: Many thanks for the suggestion. Please find below the Reviewer's token to access the data submitted to GEO.

GSE131665: ajsjkccsnxglxyh

GSE131669: izsdqsumzvezfsv

GSE131424: mjingusqrzmtndon

GSE131431: itmzeucqbpudpwt

2. Line 204: The authors should refrain from conclusion regarding TE length without doing the corresponding TE length analysis.

Reply: We have now removed the sentence as it is irreverent to the analysis presented.

3. The title mentions epigenetic reprogramming but realistically the authors have only investigated DNA methylation and I strongly suggest to change the title to include 'Methylation' reprogramming.

Reply: Many thanks for the suggestion. We have revised the title of the MS as suggested.

Reviewer #3 (Remarks to the Author):

The authors studied methylation changes during seed development in chickpea large-seeded cultivar (JGK 3) and small-seeded cultivar (Himchana 1). A comparison of DNA methylomes between JGK 3 and a small-seeded cultivar (Himchana 1) revealed that 53 candidate genes involved in seed size/weight determination exhibiting hypermethylation in CG context within gene body and higher expression in JGK 3. The authors concluded that DNA methylation mediated regulation of seed development and seed size/weight determination in chickpea. This is an interesting conclusion. However, there are several concerns.

Reply: Many thanks for comprehensive review of the MS and important suggestions.

The first major concern is that the current analysis result cannot fully support the statement. The major conclusion of this manuscript is, as shown in the abstract, "Overall, this study provides insights into DNA methylation mediated regulation of seed development and seed size/weight determination in chickpea." However, the data cannot support the statement as indicated in the comment (2) and (3). The overstatement includes the following statements.

(1) Line 232:

The expression levels of genes associated with methylated intragenic TEs was significantly lower than the genes associated with non-methylated intragenic TEs except at S2 stage (Fig. 6a).

=>

(i) The authors did not examine if these TEs are located in intron or exon.

(ii) The authors did not go deeper to examine if these genes are TE-like genes. Genome annotation may annotate TE remnants as genes, because the flanking repeat sequences experienced heavy mutation leading to not being able to be predicted as TEs and the middle TE proteins are still predicted by gene prediction program. As a result, some genes are "TE genes" derived from TEs.

(iii) The authors did not go deeper to examine if these genes are pseudogenes with specific criteria like ENCODE project with specific criteris to define a pseudo gene.

(iv) When examining the expression profile in Fig 6(a), 75% of the genes with methylated TEs in gene bodies are silent except S2. These group of genes could be a mixtures of real functional genes, pseudogenes and TE-like genes.

(v) Without deeper analysis indicated above, the current analysis cannot support the statement that " The expression levels of genes associated with methylated intragenic TEs was significantly lower than the genes associated with non-methylated intragenic TEs except at S2 stage".

Reply: We re-analyzed the expression levels of genes harboring methylated and non-methylated TEs in CG, CHG and CHH contexts within gene body and flanking regions with revised criteria. Now the results are similar for all the stages (including S2 stage) of seed development. As we can see from the revised analysis, most of the genes harboring methylated and non-methylated TEs within gene body are expressed. Further, it may be noted that same set of chickpea genes (annotated in the chickpea genome) have been analyzed in many other studies from our and other groups and a larger fraction of them was found to be expressed in different biological contexts. Thus, the genome annotation seems to be quite good.

(2) Line 329

...suggesting important role of intragenic TEs in seed size/weight determination in these sets of genes

=>

It is overstatement. The result just shows that, using grain filling DEGs as example, grain filling DEGs can be divided into two groups, (i) some are not differentially methylated, and, (ii) some are differentially methylated. That's all. The authors may examine that in the grain filling DGEs, the gene number of DMRs-associated ones are statistically significantly higher than the ones with no DMR association.

Reply: Many thanks for the suggestion. We have now provided number of DEGs associated and not associated with DMRs for different sets of genes (Supplementary Fig. 16a). The said statement has been moderated to reflect the results, as suggested.

(3) Line 358

These results suggest that DNA methylation play an important role in regulating candidate genes involved in seed size/weight determination in chickpea cultivar(s).

=>

(i) It is known that gene with medium high expression levels are CG-gene body methylated (Nat Genet. 2007;39:61-9).

(ii) In bigger seeds, relative genes sets (grain filling etc) are higher expressed than in small seeds and they are DEGs.

(iii) Thus, it is not surprised that these grain filling genes have gene body methylation in big seeds.

(iv) There are many possibilities there are CG gene body methylation in genes with medium high RNA levels and the role is still enigmatic (Curr Opin Plant Biol. 2017;36:103-110). Also one possibility is that the CG gene body methylation occurs in these highly expressed genes relative to big seeds phenotypes in order to prevent spurious transcription initiation (Nature. 2017;543:72-77).

(v) Together, the authors have no sufficient data to fully support the statement that “DNA methylation play an important role in regulating candidate genes involved in seed size/weight determination”.

Reply: We have moderated the said statement in the revised MS. It may be noted that some of the known facts indicated by the Reviewer are already mentioned/cited in the “Discussion” section of the MS. We added few more statements reflecting the facts, as suggested.

Second, there is no definition of many terms used in this manuscript leading to not possible to judge if the analysis result can fully support the statement. This includes the following statements.

(4) Line 143:

Methylation dynamics and its impact on differential gene expression during seed development => What are definition of promoter regions and downstream regions in this analysis in terms of distance to transcription start site and termination site? Besides, if one DMR is located across promoter and gene body, then the this is counted as promoter or gene body? Same as downstream and gene body. Please clarify it.

Reply: We have added more details in the “Methods” and “Results” Sections at relevant places in the revised MS for clarification.

(5) Line 288

The number of hypomethylated DMRs were marginally higher than hypermethylated DMRs in JGK 3...

=> No definition the DMR is derived from which cultivar minus which one. Then not possible to judge if the data can support the statement.

Reply: Many thanks for the suggestion. We have provided the desired information in the revised MS.

(6) Line 308:

Further, GO (biological process) analysis of hyper and/or hypomethylated genes showed enrichment of cell cycle,

=> There is no definition of hyper and/or hypomethylated genes in this manuscript. Is it DMR-associated genes?

Reply: Yes, the said set of genes represent the DMR-associated genes. We have now provided this definition clearly in the revised MS.

The third major concern is derived from no detailed information leading to not possible to judge if the data can fully support the statement. This includes the following statements.

(7) Line 179:

Further, we analyzed the influence of differential methylation on differential gene expression for sets of genes involved in important biological processes during seed development. We selected four sets of

genes, including those involved in cell cycle, differentiation, grain filling and desiccation processes based on their associated GO terms (Supplementary Fig. 5).

=> MAIN CONCERNS: the statement is the role of DNA methylation in regulating important sets of genes involved in important biological processes during seed development. However, it is not clear what genes are exactly tested in this analysis.

(i) What are genes used in the analysis to get the result of Supplemental Fig 5? The authors should show the gene list.

(ii) What's the criteria to choose genes in this analysis? It is based on the reference papers? If yes, please list the reference papers.

Reply: We have provided the desired details in the “Methods” section and provided the list of sets of genes used in the study as Supplementary Dataset S2.

(8) Line 240 to line 248

MAJOR CONCERNS: No sufficient information and analysis to judge if the analysis result can support the statement or not.

DMR associated genes harboured significantly higher frequency of TEs within their gene body during all the stage transitions (Fig. 6b)...

=>

(i) The DMRs here should be divided into CG, CHG and CHH contexts to reveal more information. If there are many CHG and CHH DMRs here, then it may suggest these DMRs are derived from TEs. If it is true that there are many CHG and CHH DMRs, then it is not surprised that DMR associated genes have higher number of TEs as compared non-DMR associated genes.

(ii) In Fig 6(b) and SFig 7 (b) and (c), the authors did statistic test here. The authors should show the original data of the what four numbers here were used for doing Fischer exact test as supplemental table. Thus, the readers have better idea how the authors did analysis to get the result.

Reply: The frequency of TEs within gene body and/or its flanking regions has been reanalyzed in CG, CHG and CHH contexts, as suggested. The level of significance (*p*-value) was determined using Wilcoxon Signed rank test and has been mentioned in the legends.

The forth major concern is that the statement made by analysis which has no appropriate control as below.

(9) Line 251:

Interestingly, frequency of intragenic TEs was significantly higher in the DMR-associated DEGs in all the four sets of genes analyzed too (Supplementary Fig. 7)

=>

The authors need a control for this analysis: doing the same analysis in DMR associated genes in promoter and downstream regions. Without the control, the conclusion (intragenic TEs play important role in regulating gene regulation during seed development) is suspicious.

Reply: Many thanks for the suggestion. We compared the frequency of TEs within gene body and flanking regions for the four sets of DMR associated DEGs in Supplementary Fig. 9.

The following is the “novelty” issue.

In abstract:

(10) Higher frequency of small RNAs in hypermethylated TEs in successive stages suggested role of RNA-dependent DNA methylation (RdDM) pathway.

=> Annu Rev Plant Biol. 2015;66:243-67.

Reply: We have moderated the sentence now.

(11) Progressive gain of DNA methylation in CHH context in transposable elements (TEs) was observed during seed development.

=> Proc Natl Acad Sci U S A. 2017 Nov 7;114(45):E9730-E9739.

Reply: We have moderated the sentence now.

(12) In The 2nd result section: Influence of DNA methylation on gene expression during seed development

=> It's been known in Nat Genet. 2007 Jan;39(1):61-9.

Reply: We agree with the Reviewer's comment. However, the pattern may differ in different plants and biological contexts. Thus, it is important to analyze the same in the given biological context in chickpea too. We have now revised the text at relevant places. The said reference has also been cited in the MS.

(13) The forth result section: Epigenetic reprogramming of TEs during seed development
The conclusion is "These results suggest that RdDM-independent pathway also complement RdDM dependent pathway to some extent in TE methylation during seed development".

=> It's been known in Cell. 2013 Mar 28;153(1):193-205.

Reply: The reference has been cited in the MS.

The following is the "statistic test" issue.

(14) Line 157:

The methylation level differences were more significant in CG and CHG contexts during S3/S5 and S5/S7 transitions (Fig. 2c).

=> What is statistic test and cutoff p value in this analysis?

Reply: Many thanks for the suggestion. We found a larger fraction of genes with higher methylation level (%) difference in CG and CHG contexts during S3/S5 and S5/S7 transitions (earlier Fig. 2c). We have moderated the statement. In addition, we have now moved this figure to the Supplementary Material (Supplementary Fig. 5).

(15) Line 172:

Significant correlation was observed in specific sequence context(s) during specific stage transitions

=> What statistic test was done here?

Reply: Many thanks for the suggestion. Here, we have analyzed the trend between direction of differential methylation and differential expression. Thus, we have moderated the statement.

The following is minor concerns in introduction.

(16) Line 40

Epigenetic modifications control reorganization of chromatin structures to determine euchromatic or heterochromatic regions driven by internal and/or environmental cues

=> need to add reference

Reply: Relevant references have been added.

The followings are minor concerns in results.

(17) Line 96: “We analyzed early-embryogenesis (S1), mid-embryogenesis (S2), late-embryogenesis (S3), mid-maturation (S5) and late maturation (S7) stages”

=> Add seed photo as Fig 1 in Plant J. 91(6), 1088–110 (2017) to help the readers have better idea what they are.

Reply: Many thanks for the suggestion. We added photographs of the seed stages analyzed in the study.

(18) Line 103:

“only 0.006% read pairs mapped to the 104 chloroplast genome, which confirmed high efficiency of bisulphite conversion in our 105 experiments (Supplementary Table 1).”

=> Is it the un-conversion rate? The way to calculate un-conversion rate is to compare the reference genome sequence vs BS-seq sequence at base resolution, rather than “reads map the chloroplast genome”. Please clarify it.

Reply: Yes, it is non-conversion rate. As the chloroplast genome is considered to be naturally unmethylated, the mapping against chloroplast genome give a fair idea about the non-conversion rate. This is a standard method being followed in most of the publications related to whole genome DNA methylation analysis in plants. We have moderated the earlier sentence and added more details for better clarification on this aspect.

(19) Line 120:

...context was observed during seed development (Fig. 1c). The gain of methylation was much higher at the S7 stage especially in the distal flanking regions (>500 bp from genic ends).
=> No scale on the x-axis leading to not being possible to determine the location of 500 bp. Please add the 500 bp position on the x-axis of the figure.

Reply: In the revised metaplots, we do not observe the drastic higher methylation level in CHH context in the distal flanking regions. Therefore, we removed the said sentence.

(20) Line 170:

At global level, no consistent correlation pattern between differential methylation and differential gene expression at all the successive stages of seed development, was observed...

=> There are so many numbers in the fig 3b and 3c. It is very confusing what the authors exactly mean “At global level”?

Reply: We are sorry for the confusion. We have revised the sentence for clarification.

(21) Line 177:

However, the number of genes showing correlation represented only a minor fraction of the total DMR-associated DEGs...

=>

(i) Does this “correlation” means positive correlation only? Or negative correlation? Or together?

(ii) What does this mean exactly? Does this mean “significant correlation was observed in specific sequence context(s) during specific stage transitions” is only a minor fraction? If using statistic test to examine the fraction of genes are “significant correlation was observed in specific sequence context(s) during specific stage transitions” to conclude that “the number of genes showing correlation represented only a minor fraction of the total DMR-associated DEGs”, then the authors should make a summary table to show the fraction of genes which pass the statistic test cutoff value.

Reply: Many thanks for the suggestion. We used “correlation” reluctantly to reflect the relationship between direction of differential methylation and differential expression in the earlier version. We have moderated the statements now for more clarity.

(22) Line 286

Further, we identified DMR-associated DEGs between the two cultivars (JGK3/Himchana1) at S3 and S5 stages of seed development

=>

(i) It is DMR-associated “genes” or “DEGs”? Should be “genes”, right? Because in line 315, you indicated “we identified DMR-associated DEGs between the cultivars” and talked about DMR-associated DEG for the entire paragraph after line 315. But in Fig 7d for this paragraph, you indicated it is “genes”. This error occurs in the critical sentence makes the readers very confusing which cripples the readers in terms of understating the content.

(ii) it is very confusing what is the definition of the DMRs-associated genes? DMRs in promoter? Gene body? Downstream regions?

(iii) Not sure the logic flow for the entire paragraph. This sentence saying “DMR-associated DEGs” were identified, but the following entire paragraph is nothing relative to DEGs, but DMRs and DMR-associated genes only.

Reply: Many thanks for the suggestion and we are very sorry for the confusion due to inadvertent mistake. Yes, we agree that it should be genes. We revised the sentence as suggested. We have defined the DMR-associated genes in the revised MS. The description of the entire paragraph is majorly for DMR-associated genes. We have made the desired corrections in it.

(23) Line 287

higher fraction of DMRs were detected in CG context at both the stages of seed development
=> Higher than what? What to compare?

Reply: We corrected the paragraph in the revised MS.

(24) Line 290

9734 hypomethylated DMRs

=> 9734 here, but 9724 in the fig 7a. Please clarify it.

Reply: Many thanks for pointing out the mistake. It has been corrected to 9724.

The followings are minor concerns in figures and tables.

(25) Fig. 1b

=> Not clear how to get the boxplot. Is it the single C site? Or the sliding window with specific size?

Reply: It is for single mC site. The description has been added in the figure legend.

(26) Fig. 3b, c:

=> (i) Between Fig 3a and 3b, the number of genes cannot match with each other. For example, in fig 3a, 382 genes in S1/S2 are DMR-associated genes and DEGs. However, in fig 3b, the sum of the gene numbers in S1/S2 is higher than 382.

(ii) Besides, what is the color intensity of green and red boxes?

Reply: The higher number of DMRs in Fig. 3b than Fig. 3a is due to occurrence of same DMRs in more than one sequence context and/or gene regions in some cases. Color intensity of red and green indicates number of upregulated and downregulated genes, respectively. The details have been added in the revised MS.

(27) Fig. 4b:

=> Fig 4b: the promoter of TE is in the repeat ends, not outside the TE. Thus, the “promoter” on the fig 4b should be corrected.

Reply: We have changed the “promoter” to “upstream” at all places.

(28) Fig. 6a:

It is suspicious that that the expression levels of genes with methylated TEs insertion in gene bodies are as similar to genes with non-methylated TEs insertion in gene bodies. The authors may check it again.

Reply: We reanalyzed the expression levels of genes harboring methylated and/or non-methylated TEs in different sequence contexts and gene regions. The ambiguity has been resolved now.

(29) Supplementary Table 2:

=> The small RNA number in Stab 2 is the normalized result or not? Please clarify it.

Reply: Many thanks for the suggestion. The number of small RNAs given in Supplementary Table 2 is the number of unique small RNAs after pre-processing of the raw data. The details have been mentioned in the revised MS.

(30) Supplementary Fig. 2

=> Not clear how to get the boxplot. Is it the single C site? Or the sliding window with specific size?

Reply: The boxplot has been generated using the methylation level at individual mC sites. The description has been added in the figure legend.

(31) Supplementary Fig. 3

=> Please add label of S2 –S7 on the fig.

Reply: We added labels S2-S7 in the Supplementary Fig. 3, as suggested.

(32) Supplementary Fig. 4:

=> The heat map does not match the corresponding GO terms on the right part of the heatmap.

Reply: This seems to have happened during PDF conversion of the MS. We ensured to upload the correct version this time. Please note that the said Figure has been moved as Fig. 2c.

(33) Supplementary Fig. 10

=> Fig on the left does not fit the table on the right.

Reply: This seems to have happened during PDF conversion of the MS. We ensured to upload the correct version this time. Please note that the said Figure has been moved as Fig. 7f.

(34) Supplementary Fig. 11

=> The heat map figure is twisted.

Reply: This seems to have happened during PDF conversion of the MS. We ensured to upload the correct version this time.

In summary, data in this this manuscript cannot support some statements, especially the major conclusion, “Overall, this study provides insights into DNA methylation mediated regulation of seed development and seed size/weight determination in chickpea.” Furthermore, missing detailed information and well-defined terms used in this manuscript results in not possible to judge if the analysis results can support the statements. On the other hand, some statements are not novel. Finally, this manuscript has many editing errors. I would suggest the authors to re-think about the analysis results and make the statement carefully.

Reply: Many thanks for a thorough review and constructive comments/suggestions. We have now revised the manuscript considering all the comments. In addition, we made a sincere effort to edit the whole MS for any other correction(s).

REVIEWERS' COMMENTS:

Reviewer #1 (Remarks to the Author):

I appreciate the substantial work the authors have done to address my comments. The manuscript is greatly improved and I am in favor of publication.

Reviewer #2 (Remarks to the Author):

The authors have satisfactorily addressed all my concerns. I thank the authors for the additional analyses and revisions to the manuscript.